# Olfactory ensheathing cells from adult female rats are hybrid glia that promote neural repair

Patricia E Phelps*, Sung Min Ha, Rana R Khankan, Mahlet A Mekonnen, Giovanni Juarez, Kaitlin L Ingraham Dixie, Yen-Wei Chen, Xia Yang*

Department of Integrative Biology and Physiology, UCLA, Los Angeles, United States

## eLife Assessment

This study unveils **important** data describing cell states of olfactory ensheathing cells, and how these cell states may relate to repair after spinal cord injury. The framework used for characterizing these cells is **solid**. This work will be of interest to stem cell biologists and spinal cord injury researchers.

*For correspondence:
pphelps@physci.ucla.edu (PEP);
xyang123@ucla.edu (XY)

**Competing interest:** The authors declare that no competing interests exist.

**Abstract** Olfactory ensheathing cells (OECs) are unique glial cells found in both central and peripheral nervous systems where they support continuous axonal outgrowth of olfactory sensory neurons to their targets. Previously, we reported that following severe spinal cord injury, OECs transplanted near the injury site modify the inhibitory glial scar and facilitate axon regeneration past the scar border and into the lesion. To better understand the mechanisms underlying the reparative properties of OECs, we used single-cell RNA-sequencing of OECs from adult rats to study their gene expression programs. Our analyses revealed five diverse OEC subtypes, each expressing novel marker genes and pathways indicative of progenitor, axonal regeneration, secreted molecules, or microglia-like functions. We found substantial overlap of OEC genes with those of Schwann cells, but also with microglia, astrocytes, and oligodendrocytes. We confirmed established markers on cultured OECs, and localized select top genes of OEC subtypes in olfactory bulb tissue. We also show that OECs secrete Reelin and Connective tissue growth factor, extracellular matrix molecules which are important for neural repair and axonal outgrowth. Our results support that OECs are a unique hybrid glia, some with progenitor characteristics, and that their gene expression patterns indicate functions related to wound healing, injury repair, and axonal regeneration.

## Introduction

Olfactory sensory neurons (OSNs) are remarkable as they undergo life-long neurogenesis within the adult olfactory epithelium (*Graziadei and Monti Graziadei, 1985*), and their axons are exclusively associated with a specialized type of glia, olfactory ensheathing cells (OECs; *Doucette, 1990*). OECs provide the outgrowth and guidance factors needed for OSN axons to reach their glomerular targets and exhibit regenerative abilities in response to injury (*Doucette, 1991*; *Li et al., 2005*; *Williams et al., 2004*). OECs have received considerable interest due to their ability to support axonal outgrowth even after severe or complete spinal cord transection in rodents, dogs, and humans (*Granger et al., 2012*; *Khankan et al., 2016*; *Ramón-Cueto et al., 2000*; *Tabakow et al., 2014*; *Takeoka et al., 2011*; *Thornton et al., 2018*).

The regenerative capacity of adult spinal cord neurons is inhibited by the lesion site environment that includes a reactive glial scar, together with invading meningeal fibroblasts and multiple immune cells

(*Burda and Sofroniew, 2014*; *Wanner et al., 2013*). The lesion core is composed of non-neural tissue that inhibits axonal outgrowth due to the upregulation and secretion of chondroitin sulfate proteoglycans that impede axon regeneration (*Cregg et al., 2014*). Some studies of severe or complete spinal cord transection in rats followed by the transplantation of OECs provided evidence that OECs surround axon bundles which regenerate into and occasionally across the injury site (*Khankan et al., 2016*; *Ramón-Cueto et al., 2000*; *Takeoka et al., 2011*; *Thornton et al., 2018*). In combination with in vitro studies, detailed analyses of injury sites suggest that OECs: (1) function as phagocytes to clear degenerating axonal debris and necrotic cells (*Khankan et al., 2016*; *Nazareth et al., 2020*; *Su et al., 2013*), (2) modulate the immune response to injury (*Khankan et al., 2016*; *Vincent et al., 2007*), and (3) interact favorably with the glial scar and lesion core that form after injury in vitro (*Lakatos et al., 2003*; *Lakatos et al., 2000*) and in vivo (*Khankan et al., 2016*; *Thornton et al., 2018*). OEC transplantation following complete spinal cord transection creates an environment that is conducive to neural repair (*Khankan et al., 2016*; *Takeoka et al., 2011*; *Thornton et al., 2018*). OECs secrete molecules that stimulate axon outgrowth such as Brain-derived neurotrophic factor (Bdnf), Nerve growth factor (Ngf), and Laminin (*Ruitenberg et al., 2003*; *Runyan and Phelps, 2009*; *Woodhall et al., 2001*) and enhance cell-to-cell-mediated interactions between neurites and OECs in an inhibitory environment (*Chung et al., 2004*; *Khankan et al., 2015*; *Windus et al., 2007*). *Khankan et al., 2015* showed that neurites grew two to three times longer in growth-inhibitory areas of multicellular 'scar-like' cultures when they were directly associated with olfactory bulb-derived OECs (OB-OECs), but the molecular mechanisms are unknown.

The outcomes of OEC transplantation studies after spinal cord injury (SCI) vary substantially in the literature due to many technical differences in their experimental designs. The source of OECs has a great impact on the outcome, with OB-OECs showing more promise than lamina propria-derived OECs, and purified, freshly prepared OECs being required for optimal OEC survival. Other important variables include the severity of the injury (hemisection to complete spinal cord transection), the age of the spinal cord injured host (early postnatal vs. adult), and OEC transplant strategies (delayed or acute transplantation, cell transplants with or without a matrix; *Franssen et al., 2007*). *Franssen et al., 2007* evaluated OEC transplantation studies, and reported that 41 out of 56 studies showed positive effects, such as stimulation of regeneration, positive interactions with the glial scar and remyelination of axons. More recent reviews and meta-analyses of the effects of OEC transplantation following different SCI models reported that OECs significantly improved locomotor function (*Watzlawick et al., 2016*; *Nakhjavan-Shahraki et al., 2018*).

Together with collaborators, we conducted six SCI studies in adult rats with a completely transected, thoracic spinal cord model followed by OB-OEC transplantation (*Kubasak et al., 2008*; *Takeoka et al., 2011*; *Khankan et al., 2016*; *Thornton et al., 2018*; *Dixie, 2019*). Results from five of the six studies showed physiological and/or anatomical evidence of axonal regeneration into and occasionally across the injury site. In 6- to 8-month-long studies, *Takeoka et al., 2011* reported physiological evidence of motor connectivity across the transection in OEC- but not media-transplanted rats. These experiments used transcranial electric stimulation of the motor cortex or brainstem to detect motor-evoked potentials (MEPs) with EMG electrodes in hindlimb muscles at 4- and 7-month post-transection. After 7 months, 70% of OEC-treated rats responded to stimulation with hindlimb MEPs (motor cortex, 5/20; brainstem 12/20; *Takeoka et al., 2011*). A complete re-transection above the original transection was carried out 1 month later and all MEPs in OEC-injected rats were eliminated. These results provide physiological evidence of axon conductivity across the injury site in OEC-treated rats. Additionally, three of our long-term studies evaluated anatomical axonal outgrowth of the descending serotonergic Raphespinal pathway into and through the injury site. Significantly more serotonergic-labeled axons crossed the rostral inhibitory scar border (*Takeoka et al., 2011*) or occupied a larger area within the injury site core (*Thornton et al., 2018*; *Dixie, 2019*) in OEC-transplanted rats than in fibroblast or media controls. In addition, significantly more neurofilament-labeled axons were found within the lesion core of OEC-transplanted versus control rats (*Thornton et al., 2018*; *Dixie, 2019*).

The question addressed in the present study is how OB-OECs perform the diverse functions associated with neural repair. We hypothesize that there exist OB-OEC subtypes, and that their respective gene programs and secreted molecules underlie their multifaceted reparative activities. Here, we identify the OB-OEC subtypes and their molecular programs that contribute to injury repair, such as

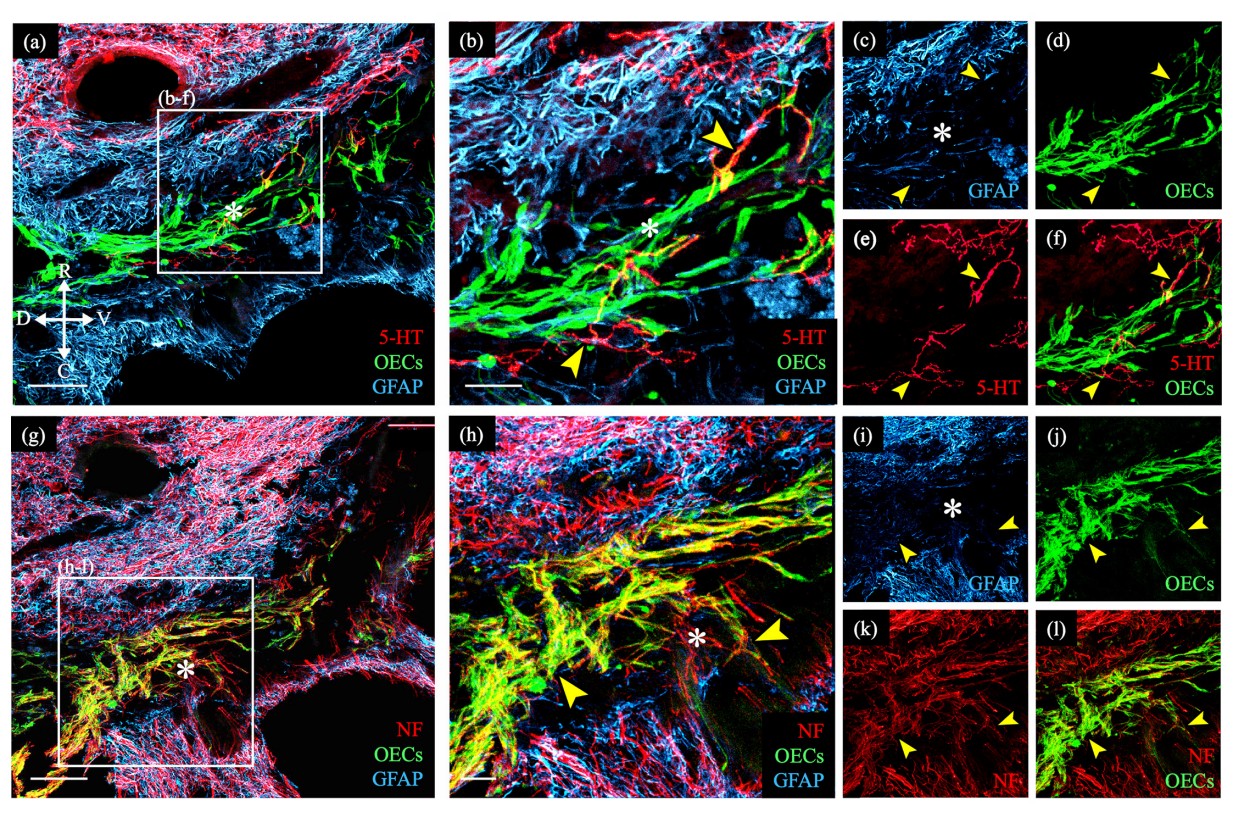

**Figure 1.** Transplanted GFP-OECs in the center of the lesion core associate with numerous axons. Sagittal sections show rostral and caudal glial scar borders of the injury site which are identified with glial fibrillary acidic protein (GFAP, blue). The GFAP-negative lesion core contains GFP-OECs (green) and is marked with asterisks. Arrowheads mark axons crossing into the lesion core. (**a–f**) Serotonergic axons (5-HT, red) are found in the rostral spinal cord stump and associate with olfactory ensheathing cells (OECs) (green) in the lesion core. Single channels for GFAP (**c**), OECs (**d**), 5-HT (**e**), and a combination of 5-HT and OECs (**f**). (**g–l**) Nearby injury site section from the same rat (**a**). Numerous neurofilament-positive axons (white) are associated with the OECs (green) in the lesion core. Single channels for GFAP (**i**), OECs (**j**), neurofilament (**k**), and a combination of neurofilament and OECs (**l**). Bridge formation across the injury site was observed in 2 of 8 OEC-transplanted and 0 of 11 fibroblast- or media-transplanted, spinal cord transected rats. Combined with the 1/5 OEC-transplanted rats with axons crossing the injury and 0/5 fibroblast controls in our previous study (*Thornton et al., 2018*), we observed bridges in 3/13 OEC-transplanted rats versus 0/16 controls (p = 0.042, two-sample proportion test). Scale bars: a, g = 500 µm; b, h = 100 µm. Reprinted from *Dixie, 2019*, *UCLA*. ProQuest ID: Dixie_ucla_0031D_18445.

the growth-stimulating secreted and cell adhesion molecules involved in OB-OEC interactions that promote neurite and axonal outgrowth. We used single-cell RNA-sequencing (scRNA-seq) to characterize the immunopurified female rat OB-OECs (hereafter called OECs unless stated otherwise) that are similar to those transplanted into our previous female rat SCI studies. We first compared the genes expressed by purified OECs with those expressed by the 'leftover cells', that is, the cells which are not selected by the panning procedure, to determine if the purification process selected for specific OEC subtypes. After confirming the characteristic OEC markers, our scRNA-seq data revealed five OEC subtypes which expressed unique marker genes and were further characterized and confirmed experimentally. Finally, we examined interesting extracellular matrix (ECM) molecules secreted by OECs, Reelin (*Reln*) and Connective tissue growth factor (*Ccn2/Ctgf*), both of which may facilitate axonal outgrowth into the inhibitory injury core environment.

## Results

One particularly challenging aspect facing neural repair of SCI is to facilitate axonal outgrowth and eventually regeneration across the inhibitory injury site. A common finding in our studies testing the effects of OEC transplantation following SCI is that OECs modify the injury site so that some axons project past the glial scar and into the lesion core surrounded by OECs (*Dixie, 2019*; *Khankan et al.,*

*2016*; *Takeoka et al., 2011*; *Thornton et al., 2018*). *Figure 1* shows an enlargement of an injury site which contains green fluorescent protein (GFP)-labeled OECs in the glial fibrillary acidic protein (GFAP)-negative injury core between the two GFAP-positive spinal cord stumps. The OECs in the injury core are associated with a few serotonergic- (*Figure 1a–f*) and many neurofilament-labeled axons (*Figure 1g–l*). We note, however, that such bridge formation is rare following severe SCI in adult mammals and was detected in 2 out of 8 OEC-transplanted rats versus 0/11 media or fibroblast-transplanted controls in this study (*Dixie, 2019*). Combined with the 1/5 OEC-transplanted rats with axons crossing the injury and 0/5 fibroblast controls in our previous study (*Thornton et al., 2018*), we observed bridges in 3/13 OEC-transplanted rats versus 0/16 controls (p = 0.042, two-sample proportion test). Bridge formation, in conjunction with the additional physiological and anatomical evidence of axonal connections across the injury site presented in our previous studies, strongly supports the capacity of OECs in neural repair. To learn more about possible mechanisms by which OECs support axon regeneration, we prepared immunopurified OECs and examined their gene expression and diversity with scRNA-seq.

## Unbiased scRNA-seq of immunopurified OECs and 'leftover' controls distinguishes OECs, microglia, and fibroblasts

Using scRNA-seq, we sequenced a total of 65,481 cells across 7 samples (*n* = 3 OEC preparations and 4 'leftover' controls) that passed quality control, with an average of 9354 cells per sample. Cell clusters were visualized using t-distributed stochastic neighbor embedding (tSNE). The tSNE plot showed that cells from immunopanned OEC samples were separated from the cells from the leftover samples (*Figure 2a*). Cell clustering analysis defined a total of eight clusters (*Figure 2b*), each of which showed expression patterns of marker genes that distinguish one cluster from the others (*Figure 2c*). Based on previously reported cell type marker genes for fibroblasts and major glial cell types including OECs, astrocytes, oligodendrocytes, and microglia, we found elevated expression of OEC marker genes in clusters 2, 3, and 7, microglia marker genes in clusters 4, 6, and 7, and fibroblast marker genes in clusters 0, 1, and 5 (*Figure 2d*, *Figure 2—figure supplement 1*). In contrast, except a few select genes, the majority of markers for astrocytes and oligodendrocytes showed low expression (*Figure 2d*). The cell cluster tSNE plot with cell type labels in *Figure 2e*, and the distinct expression patterns of cell type markers are confirmed in a dot plot in *Figure 2f*. As expected, many cells enriched in the leftover controls were defined as fibroblasts and microglia based on the corresponding known marker genes, whereas cells from the immunopanned OEC samples showed high expression of OEC marker genes (*Figure 2a vs. e*). After cell type assignment, a dot plot clearly depicts distinct expression of cell-type-specific marker genes (*Figure 2f*). Our scRNA-seq results support the expected cell type composition following the immunopanning procedure. All marker genes for each major cell type, including statistics, are in *Supplementary file 1*.

We next asked if there were differences between cultures of immunopurified OECs and OECs grown in 'leftover' cultures. While our previous SCI studies transplanted 90–95% purified OECs, other labs culture and implant less purified OECs and thus many of the neighboring cells, that is, primarily fibroblasts and microglia, would be included. Our comparison showed that purified OECs express higher levels of *Cryab* (stress protection, inflammation inhibition), *Nqo1* (antioxidant protection, stress adaptation), and *Postn* (cell proliferation, survival, migration) genes (*Figure 2g*), whereas OECs in leftover cultures with fibroblasts and microglia express higher levels of *Apoe* (lipid transport and glial function), *Calcb* (inflammation and pain modulation), and *Mgp* (ECM calcium inhibitor; *Figure 2g*) than purified OECs. These results indicate immunopurified OECs may have better stress response and proliferative and survival capacity than the controls.

## Immunofluorescence verification of typical OEC marker genes revealed by scRNA-seq

To validate the OEC markers revealed by scRNA-seq, we re-cultured extra cells not used for sequencing for immunocytochemical confirmation. Because our adult OECs are immunopurified with anti-nerve growth factor receptor p75 (Ngfr[p75]), we expected and found OEC samples heavily enriched in Ngfr[p75] in both immunofluorescence (*Figure 3a*) and scRNA-seq (*Figure 3b*). Next, we detected the high expression of two common glial markers, Blbp (*Fabp7*; *Figure 3c, d*) and S100b (*Figure 3e, f*), in OECs. Sox10 is expressed in the nuclei of neural crest-derived cells such as OECs and Schwann cells,

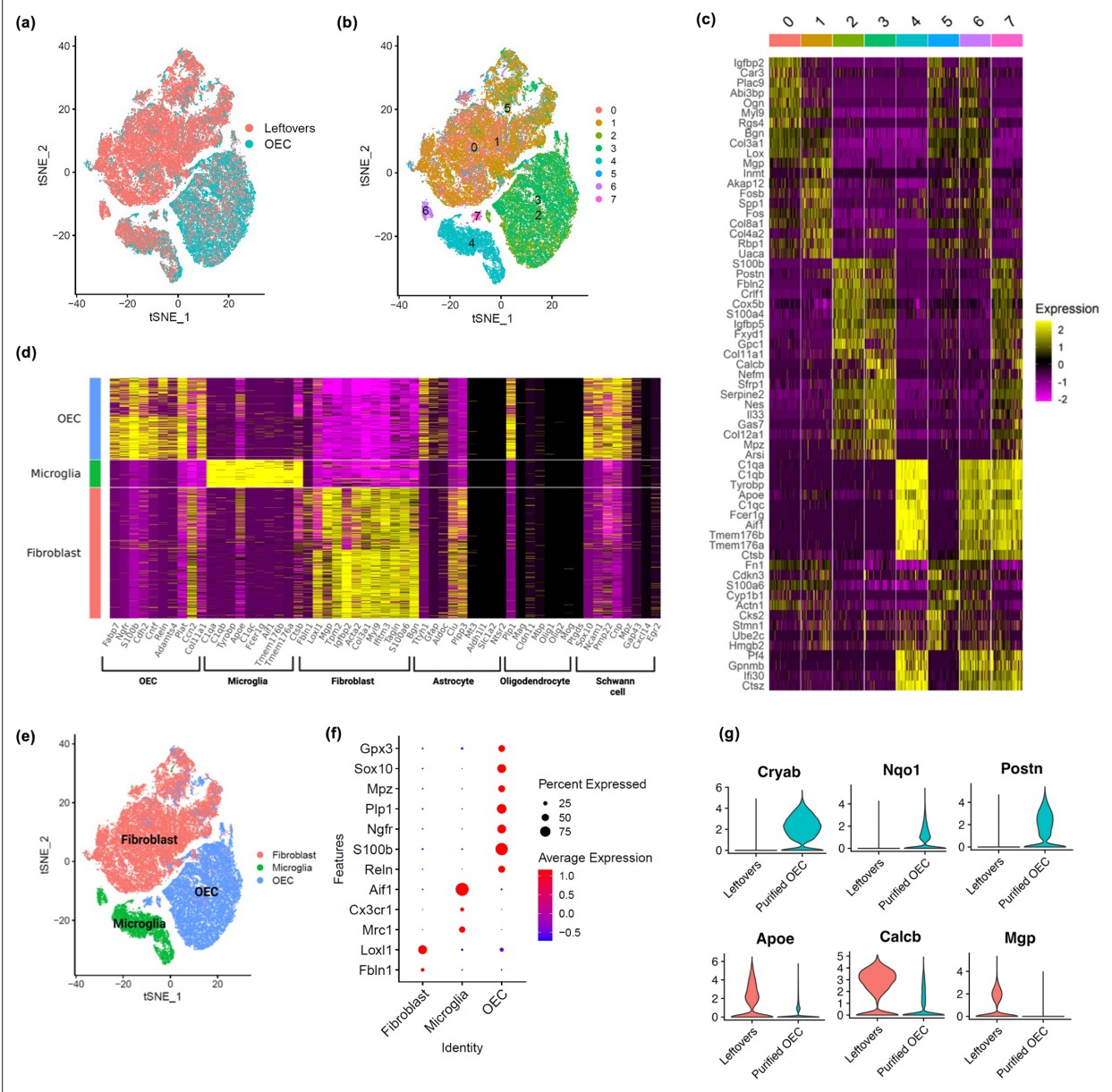

**Figure 2.** Single-cell RNA-sequencing (scRNA-seq) results show distinct clusters of olfactory ensheathing cells (OECs) and leftover cell samples. (**a**) Cells in t-distributed stochastic neighbor embedding (tSNE) plot colored by sample source, with cells from leftover samples in pink and cells from immunopurified OECs in cyan. Quality control plots in *Figure 2—figure supplement 2*. (**b**) Clustering analysis revealed a total of eight distinct cell clusters (0-7), each indicated with a different color. (**c**) A heatmap showing expression patterns of top marker genes of the eight individual cell clusters. (**d**) Cell clusters showed distinct expression patterns for known cell type markers for fibroblasts, microglia, and OECs. Clusters 0, 1, and 5 had high expression of fibroblast markers and are labeled as fibroblast in the *y*-axis; clusters 4 and 6 showed high expression of microglia markers and are labeled as microglia; clusters 2, 3, and 7 showed high expression of OEC markers and are labeled as OEC. Known marker genes for different cell types are on the *x*-axis. Feature and violin plots for select marker genes are in *Figure 2—figure supplement 1*. (**e**) Based on known cell type markers, cell clusters in tSNE plot were labeled with the corresponding cell types. (**f**) The expression of cell-type-specific marker genes is depicted in a dot plot. (**g**) Genes that distinguish purified OECs versus OECs in leftover cultures are shown. The top 3 genes were higher in purified OECs, whereas the bottom 3 genes were higher in 'leftover' cultures.

The online version of this article includes the following figure supplement(s) for figure 2:

**Figure supplement 1.** Marker genes were expressed specifically in different areas.

**Figure supplement 2.** Violin plots for quality control associated with *Figure 2a, b*.

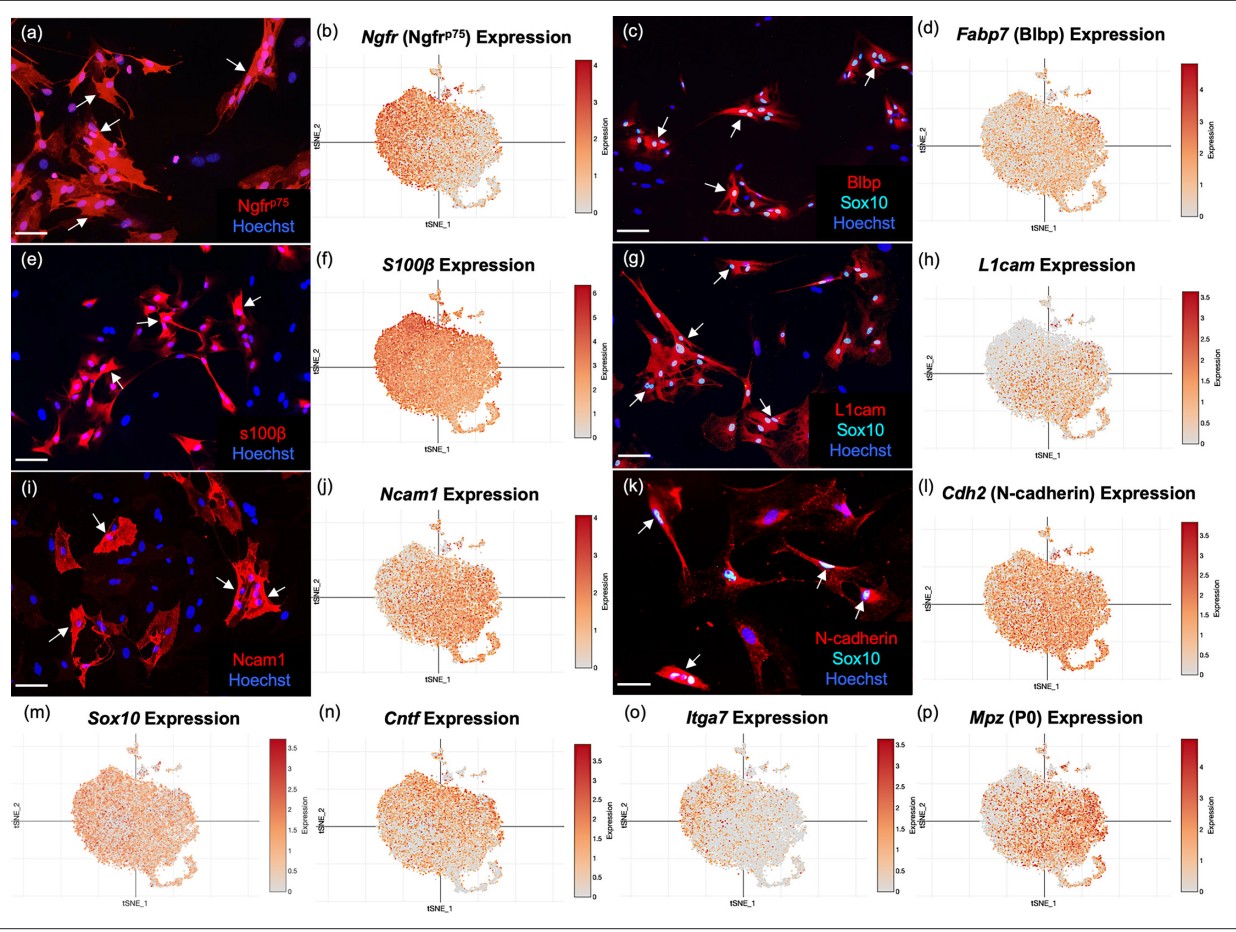

**Figure 3.** Well-established olfactory ensheathing cell (OEC) markers are revealed by single-cell RNA-sequencing (scRNA-seq) and immunofluorescence. OEC cultures were replated from extra cells prepared for scRNA-seq. Immunolabeled OECs are marked with arrows and all cell nuclei are stained with Hoechst (blue nuclei). t-Distributed stochastic neighbor embedding (tSNE) maps of the gene expression in the five clusters are shown next to the protein expression in a–l. (**a, b**) Cultured OECs express Ngfr$^{p75}$ protein and *Ngfr$^{p75}$* gene expression. (**c, d**) Blbp and Sox10 immunoreactive OECs with *Fabp7* gene expression. (**e, f**) S100β-labeled OECs together with *S100β* expression. (**g, h**) L1cam and Sox 10 labeling next to *L1cam* expression. (**i, j**) Ncam1 protein and gene expression. (**k, l**) N-Cadherin and Sox10 markers with *Cdh2* expression. (**m–p**) scRNA-seq data for *Sox10, Cntf, Itga7,* and *Mpz* (references in text). Scale bars: a, c, e, g, i, k = 50 μm.

and our cultures showed high expression levels in scRNA-seq and immunocytochemistry (*Figure 3c, g, k, m*). Three cell adhesion molecules, L1-Cam (*Figure 3g, h*; *Runyan and Phelps, 2009*; *Witheford et al., 2013*), N-Cam (*Figure 3i, j*), and N-Cadherins (*Figure 3k, l*), are also expressed in purified OEC cultures. In addition, other previously reported OEC genes are now verified by scRNA-seq, including the pro-myelinating Ciliary neurotrophic factor (*Cntf, Figure 3n*; *Asan et al., 2003*; *Roet et al., 2011*), the laminin receptor α7 integrin (*Itga* 7, *Figure 3o*; *Ingram et al., 2016*), and the myelin related gene P0 (*Mpz, Figure 3p*; *Sasaki et al., 2006*). Thus, well-characterized OEC markers were detected by scRNA-seq and most are widely distributed among OEC clusters.

In addition to confirming select genes, we also compared the 209 significant OEC markers identified from our scRNA-seq data with the 309 genes from the meta-analysis of five OEC microarray studies on cultured early-passage OB-OECs versus other tissues (*Roet et al., 2011*). We found 63 genes overlapping between our OEC markers and the published markers, representing a 15.3-fold overlap enrichment (p-value: 4.3e−58), further supporting the reliability of our scRNA-seq findings.

## Subclustering analysis of OECs revealed refined OEC subtypes and their top marker genes

Next, we extracted OECs and performed a subclustering analysis with OECs alone, and found five distinct subclusters (*Figure 4a*). All subclusters except for cluster 3 expressed previously identified

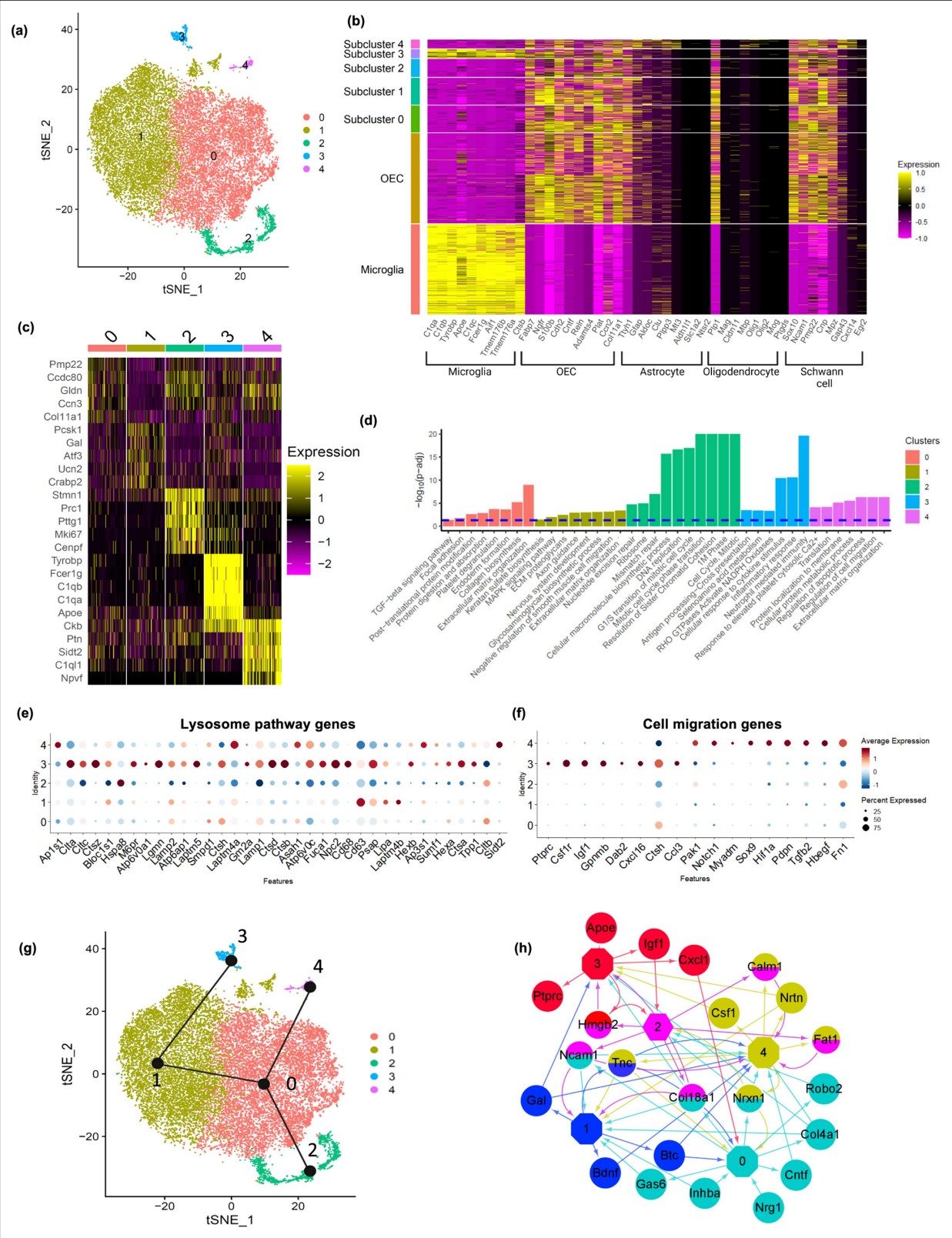

**Figure 4.** Olfactory ensheathing cell (OEC) subclusters, marker genes, and enriched pathways. (**a**) Clustering analysis of OECs revealed five separate clusters (0–4). (**b**) Heatmap depicts the expression patterns of known marker genes of glial cell types (*x*-axis) versus microglia, all OECs and OEC subclusters (*y*-axis). OEC subclusters express select markers of other glial cell types. (**c**) Heatmap depicts the top 5 marker genes (*y*-axis) for each OEC subcluster (*x*-axis). (**d**) Pathways associated with marker genes of different OEC clusters are shown. The dashed line indicates the false discovery rate

*Figure 4 continued on next page*

*Figure 4 continued*

(FDR) <0.05 in the pathway analysis. (**e**) Dot plot showing that cluster 3 has higher potential lysosomal function based on lysosome pathway genes than the other clusters. (**f**) Both clusters 3 and 4 express select genes involved in positive regulation of cell migration. (**g**) Trajectory analysis reveals two trajectories, one including subclusters 2, 0, and 4, and another involving subclusters 2, 0, 1, and 3. (**h**) NicheNet ligand–receptor network plot demonstrates intracellular communication between OEC subtypes. Hexagons represent subclusters 0–4 and circles indicate ligands secreted from subclusters. Edges (arrows) point to clusters where receptors of the ligands are expressed. Different colors represent different OEC subclusters.

markers of OECs as well as subsets of marker genes for astrocytes, oligodendrocytes, and Schwann cells (*Figure 4b, c*). The percentage of OECs within each cluster that express a gene, the percentage of OECs in all other subtypes that express a gene and their p values are found in *Supplementary file 2*. Interestingly, cluster 3 showed expression of both microglia and OEC markers. The top 20 genes expressed by each OEC subtype were investigated to evaluate potential functional differences between the subtypes (*Figure 4c*; Appendix 1), and their enriched pathways (*Figure 4d*). Most references for the specific genes within the clusters are found in Appendix 1.

## Cluster 0 is rich in matricellular proteins

Pathway enrichment analysis showed that ECM organization and collagen biosynthesis were significantly enriched in the marker genes for the largest OEC subcluster 0 (*Figure 4d*). Two of the top 10 genes are members of the Ccn family of matricellular proteins, *Ccn2/Ctgf* (Connective tissue growth factor; *Figure 5a*) and *Ccn3/NOV* (nephroblastoma overexpressed gene; *Figure 5b*, Appendix 1). Both are secreted ECM proteins which regulate the activity of growth factors and cytokines, function in wound healing, cell adhesion, and injury repair. The presence of *Ccn3* in OECs is novel, while *Ccn2/Ctgf* is well established (*Lamond and Barnett, 2013*; *Roet et al., 2011*). In *Figure 5c, c1, c2*, we show that *Ccn2* and *Sox10* are highly expressed in our cultured OECs. Because *Mokalled et al., 2016* reported that *Ctgfα* is a critical factor required for spontaneous axon regeneration following SCI site in zebrafish, we asked if GFP-labeled OECs transplanted 2 weeks following a complete spinal cord transection in adult rats, also expressed Ccn2/Ctgf. We found high levels of Ctgf expression in GFP-OECs (*n* = 4 rats) that bridged much of the injury site and also detected Ctgf on near-by cells (*Figure 5d, d1, d2*). GFP-labeled fibroblast transplantations (*n* = 3 rats) served as controls and also expressed Ctgf. Other top ECM genes are *Serpinf1* (Serine protease inhibitor), *Fbln2* (Fibulin2), *Fn1* (Fibronectin-1), and *Col11a1* (Collagen, type XI, alpha 1; Appendix 1).

The top-ranked marker gene for cluster 0 is *Pmp22* (Peripheral myelin protein 22), a tetraspan membrane glycoprotein that is highly expressed in myelinating Schwann cells, and contributes to the membrane organization of compact peripheral myelin. The third-ranked gene, *Gldn* (Gliomedin) is secreted by Schwann cells and contributes to the formation of the nodes of Ranvier. *Fst* (Follistatin) also plays a role in myelination and is expressed in areas of adult neurogenesis.

Many of the top cluster 0 genes are found in other glial cells: *Fbln2* is secreted by astrocytes, *Ccn2/Ctgf* is expressed by astrocytes and Schwann cells, *Nqo1* (NAD(P)H dehydrogenase, quinone 1 enzyme) is found in Bergmann glia, astrocytes, and oligodendrocytes, and *Marcks* (Myristoylated alanine-rich C-kinase substrate protein) regulates radial glial function and is found in astrocytes and oligodendrocytes. In addition, *Cd200* is expressed by astrocytes and oligodendrocytes, and binds to its receptor, *Cd200R*, on microglial cells. Thus, the OECs in cluster 0 express high levels of genes found in other glia, many of which are related to ECM and myelination.

## Cluster 1 contains classic OEC markers

Three of the top 5 genes in cluster 1, *Pcsk1* (proprotein convertase PC1), *Gal* (neuropeptide and member of the corticotropin-releasing factor family), and *Ucn2* (Urocortin-2) were reported in the meta-analysis by *Roet et al., 2011*. Galanin is expressed by neural progenitor cells, promotes neuronal differentiation in the subventricular zone and is involved in oligodendrocyte survival.

A well-known oligodendrocyte and Schwann gene associated with myelin, *Cnp* (2-3-cyclic nucleotide 3-phosphodiesterase), was high in OEC cluster 1 in addition to the cytokine *Il11* (Interleukin 11) that is secreted by astrocytes and enhances oligodendrocyte survival, maturation and myelin formation. The chondroitin sulfate proteoglycan Versican (*Vcan*) is another top gene expressed by astrocytes and oligodendrocyte progenitor cells.

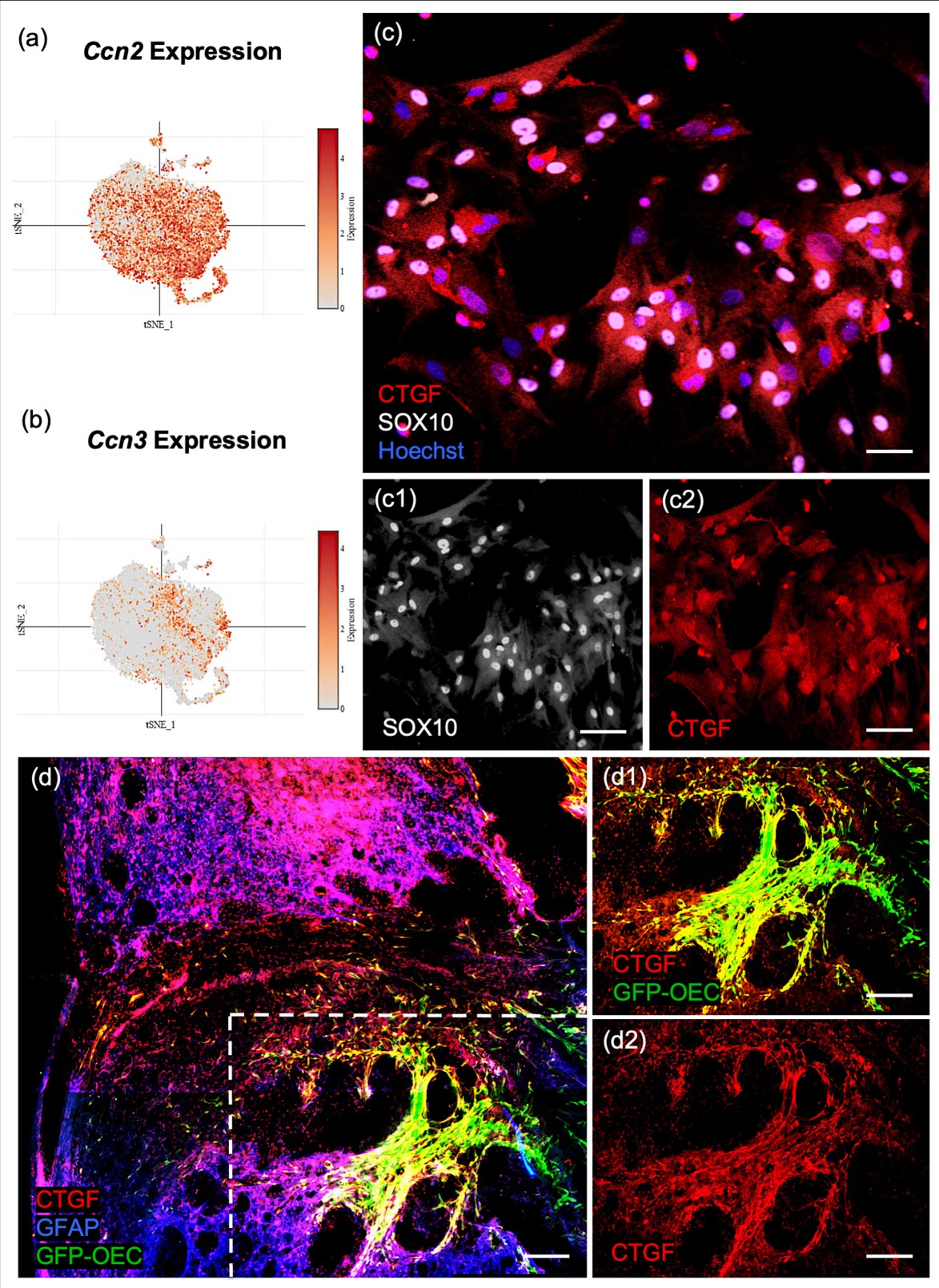

**Figure 5.** Confirmation of *Ccn2/Ctgf* (Connective tissue growth factor) in cultured olfactory ensheathing cells (OECs) and following OEC implantation after spinal cord injury. (**a, b**) The single-cell RNA-sequencing (scRNA-seq) plots of the ninth (*Ccn2/Ctgf*) and fourth (*Ccn3/Nov*) highest ranked marker genes are strongly expressed in subcluster 0. (**c, c1, c2**) The well-known matricellular protein Connective tissue growth factor (Ctgf) identifies OECs in cell cultures (red, **c, c2**), and Sox10 nuclear expression (white nuclei, **c, c1**) confirms they are neural crest-derived cells. (**d, d1, d2**) A sagittal section

*Figure 5 continued on next page*

*Figure 5 continued*

from a rat that received a complete spinal cord transection followed by green fluorescent protein (GFP)-OEC implantation was fixed 2 weeks postinjury. Glial fibrotic acidic protein (**d**, Gfap, blue) marks the edges of the borders of the glial scar. GFP-OECs (green) that express Ctgf (red) in the injury site are outlined by the box in **d**. High expression of Ctgf is detected by GFP-OECs that bridge part of the injury site in d1. The single channel of Ctgf is shown in d2. Scale bars: c–c2 = 50 μm, d–d2 = 250 μm.

Many of the genes and pathways that characterize cluster 1 are involved in nervous system development and axon regeneration (*Atf3*, *Btc*, *Gap43*, *Ngfr$^{p75}$*, *Bdnf*, and *Pcsk1*), and are found in other glia (Appendix 1). Interestingly, after nerve injury, *Atf3* (cyclic AMP-dependent transcription factor 3) is upregulated by Schwann cells in the degenerating distal nerve stump and downregulated after axon regeneration is complete (**Hunt et al., 2004**). *Btc* (Betacellulin), part of the Epidermal growth factor (Egf) family and a ligand for Egfr, also is expressed by Schwann cells after nerve injury in a pattern similar to that of *Atf3*, as is Brain-derived neurotrophic factor (*Bdnf*). In addition, the proprotein convertase PC1 (*Pcsk1*) cleaves pro-Bdnf into its active form in Schwann cells and therefore also contributes to axon outgrowth. Together, the OECs in cluster 1 secrete a number of important growth factors associated with regeneration, including a novel one, *Atf3*, reported here (Appendix 1).

## Cluster 2 represents a distinctive proliferative OEC subtype

Most top genes in cluster 2 (*Figure 6a–f*) regulate the cell cycle (*Mki67*, *Stmn1*, *Cdk1*, *Cks2*, *Cdkn3*, *Cdca8*, and *Cdca3*), are associated with mitosis (*Cenpf*, *Spc24*, *Tpx*, *Ckap2*, *Prc1*, *Ube2c*, and *Top2a*), or contribute to DNA replication and repair (*Dut* and *Fam111a*, Appendix 1). The top ranked gene, *Stmn1* (Stathmin 1; *Figure 6e*), is a microtubule destabilizing protein that is widely expressed in other OEC clusters and in areas of adult neurogenesis. Pathway enrichment confirms the proliferative property of cluster 2 with pathways such as G1/S transition of mitotic cell cycle and DNA replication

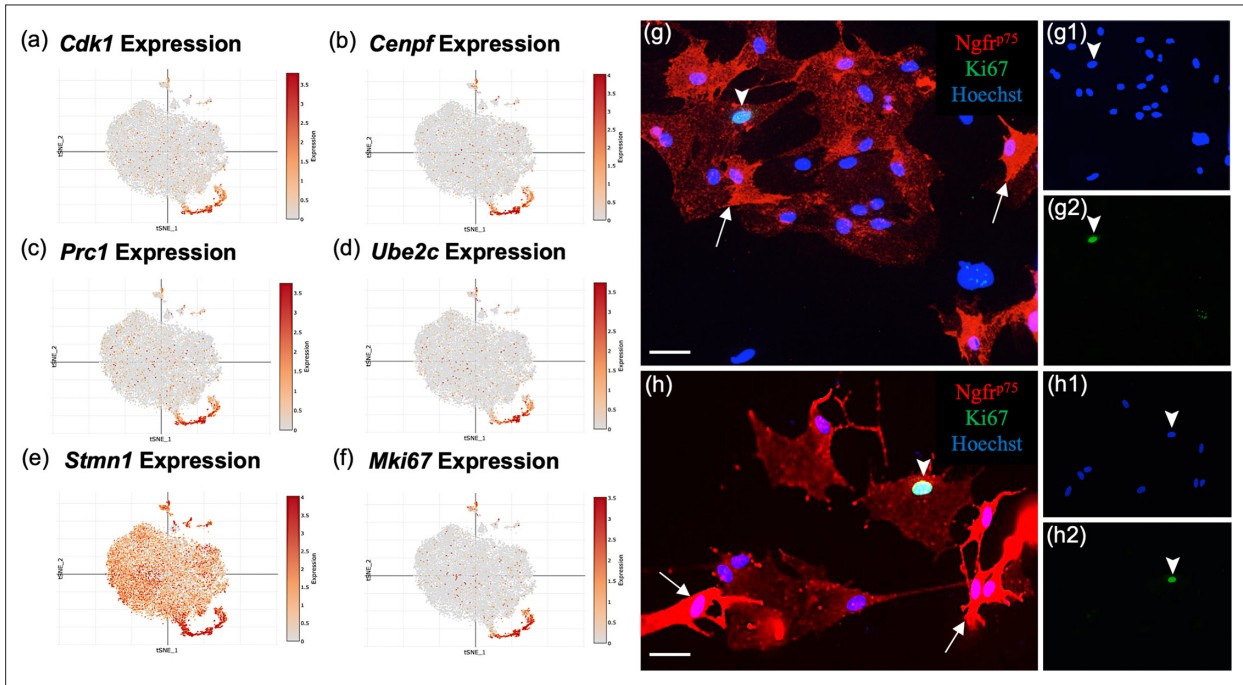

**Figure 6.** Subcluster 2 is characterized by cell cycle and proliferative markers. (**a–f**) These single-cell RNA-sequencing (scRNA-seq) plots show high expression of cell proliferation markers in cluster 2, supporting their function as olfactory ensheathing cell (OEC) progenitor cells. Only *Stmn1* (e, Stathmin, a microtubule destabilizing protein) is broadly expressed across all five clusters. (**g–g2, h–h2**) Most cultured OECs are spindle-shaped and have high Ngfr$^{p75}$ expression (red, white arrows). Of the OEC progenitors that express Ki67, 76% ± 8 of them display low levels of Ngfr$^{p75}$ immunoreactivity and a 'flat' morphology (g2, h2; green nuclei, arrowheads). The remainder of Ki67-expressing OECs express high levels of Ngfr$^{p75}$ and are fusiform in shape (24% ± 8%, n = 4 cultures, Student's t-test, p = 0.023). Hoechst marks all nuclei (g1, h1, blue nuclei, arrowheads). Scale bars: g, h = 50 μm.

(*Figure 4d*). These results show that there are distinct OEC progenitors in our cultures within the olfactory nerve layer (ONL) of the adult olfactory bulb.

*MKi67* is a well-known proliferation gene concentrated in cluster 2 (*Figure 6f*). Previous studies suggested the presence of two morphologically distinct OECs – the Schwann-cell-like, spindle-shaped OECs with high Ngfr^p75 expression and the astrocyte-like, large flat OECs with low Ngfr^p75 (*Franceschini and Barnett, 1996*). To determine if the proliferative OECs differ in appearance from adult OECs, and whether there is concordance between our OEC subtypes based on gene expression markers and previously described morphology-based OEC subtyping (*Franceschini and Barnett, 1996*), we analyzed OECs identified with the anti-Ki67 nuclear marker and anti-Ngfr^p75 (*Figure 6g, h*). Of the Ki67-positive OECs in our cultures, 24% ± 8% were strongly Ngfr^p75-positive and spindle-shaped, whereas 76% ± 8% were flat and weakly Ngfr^p75-labeled ($n$ = 4 cultures, p = 0.023). Here we show that a large percentage (~3/4^ths) of proliferative OECs are characterized by large, flat morphology and weak Ngfr^p75 expression resembling the previously described morphology-based astrocyte-like subtype. Our results indicate the two types of OEC classifications share certain degrees of overlap, indicating similarities but also differences between the two classification methods.

## Cluster 3 resembles both microglia and OECs

In addition to the typical OEC markers (*Ngfr^p75*, *S100b*, and *Sox10*), cluster 3 also expressed numerous microglia markers (*Figure 4b*). In fact, all top 20 genes in cluster 3 are expressed in microglia, macrophages, and/or monocytes (Appendix 1). Microglia markers in the top 20 genes include *Tyrobp,* which functions as an adaptor for a variety of immune receptors, *Cst3* (Cystatin C), *Anxa3* (Annexin A3), *Aif1* (Iba-1), and *Cd68*. OECs are known to endocytose bacteria and degrade axons, and therefore participate in the innate immune function (*Khankan et al., 2016*; *Leung et al., 2008*; *Nazareth et al., 2015*; *Vincent et al., 2007*). We also observed relatively high expression of genes involved in lysosome functions, such as *Cst3*, *Ctsz*, *Laptm5*, *Ctsb*, and *Lyz2*, compared to those in other OEC clusters (*Figure 4e*). Three additional top genes are involved in the complement system (*C1qA*, *C1qb*, and *C1qc*), and cluster 3 pathways are highly enriched for inflammatory response and neutrophil immunity (*Figure 4d*).

*Smithson and Kawaja, 2010* identified unique microglial/macrophages that immunolabeled with Iba-1 (*Aif1*) and Annexin A3 (*Anxa3*) in the olfactory nerve and outer nerve layer of the olfactory bulb. These authors proposed that Iba1-Anxa3 double-labeled cells were a distinct population of microglia/macrophages that protected the olfactory system against viral invasion into the cranial cavity. Based on our scRNA-seq data we offer an alternative interpretation that at least some of these Iba-1-Anxa3 cells may be a hybrid OEC-microglial cell type. Supporting this interpretation, there are a number of reports that suggest OECs frequently function as phagocytes (e.g., *Khankan et al., 2016*; *Nazareth et al., 2020*; *Su et al., 2013*).

## Cluster 4 has characteristics of astrocytes and oligodendrocytes

Cluster 4 is quite small. Its top 2 marker genes, *Sidt2*, a lysosomal membrane protein that digests macromolecules for reutilization and *Ckb*, a brain-type creatine kinase which fuels ATP-dependent cytoskeletal processes in CNS glia, are found in other OEC clusters. Four top genes, however, appear almost exclusively in cluster 4: (1) *Npvf* (Neuropeptide VF precursor/Gonadotropin-inhibitory hormone) inhibits gonadotropin secretion in several hypothalamic nuclei, and is expressed in the retina, (2) *Stmn2* (Stathmin2) is a tubulin-binding protein that regulates microtubule dynamics, (3) *Vgf* (non-acronymic) is synthesized by neurons and neuroendocrine cells and promotes oligodendrogenesis, and (4) *Mt3* (Metallothionein-3) is a small zinc-binding protein associated with growth inhibition and copper and zinc homeostasis (Appendix 1).

Other genes expressed by cluster 4 are either expressed by other glia and/or are ECM molecules. Five genes are associated with both oligodendrocytes and astrocytes (*Ckb*, *C1ql1*, *Gria2*, *Stmn2*, and *Ntrk2*, Appendix 1). Among these, the astrocytic gene *C1ql1* is a member of the Complement component 1q family and regulates synaptic connectivity by strengthening existing synapses and tagging inactive synapses for elimination. Astrocytic Thrombospondin 2 (*Thbs2*) also controls synaptogenesis and growth. Other genes reported in astrocytes include *Ptn* (Pleiotrophin), *Igfbp3* (Insulin-like growth factor-binding protein 3), and *Mt3,* whereas *Gpm6b* (Glycoprotein M6b) is involved in the formation of nodes of Ranvier in oligodendrocyte and Schwann cell myelin. Cluster 4 also has a

significant enrichment for genes involved in ECM organization (*Figure 4d*; *Actn1*, *Col81a*, *Bgn*, *Fn1*, and *Timp2*), together with clusters 0 and 1, as well as genes involved in positive regulation of cell migration together with cluster 3 (*Figure 4f*).

## Trajectory and potential interactions among OEC subclusters

We performed pseudotime trajectory analysis using the Slingshot algorithm to infer lineage trajectories, cell plasticity and lineages by ordering cells in pseudotime based on their transcriptional progression reflected in our scRNA-seq data. Transcriptional progression refers to the changes in gene expression profiles of cells as they undergo differentiation or transition through different states. The trajectory analysis results suggest that there are potential transitions between specific OEC subclusters. Our results show that there are two distinct trajectories (*Figure 4g*). The first trajectory involves clusters 2, 0, and 4, whereas the second involves clusters 2, 0, 1, and 3. Although the directionality of these trajectories is indistinguishable, the fact that cluster 2 is enriched for cell cycle genes and is the converging point of both trajectories, suggests that cluster 2 proliferates into the other OEC clusters. The predicted trajectories based on our scRNA-seq data suggest plasticity in the cell clusters.

We also modeled potential autocrine/paracrine ligand–receptor interactions between OEC subclusters using NicheNet (*Figure 4h*). This analysis revealed that clusters 0, 1, and 2 have more ligands going to other OEC clusters, whereas clusters 3 and 4 generally receive ligands from other clusters. The network also showed that clusters 0, 1, and 4 express genes encoding neurotropic factors such as Bdnf, Cntf, and Neurturin (*Ntrn*), that are involved in neuronal survival and neurite outgrowth. *Gal* from cluster 1 is a mediator for glia–glia, and glia–neuron communication (*Gresle et al., 2015*; *Ubink et al., 2003*). We found that cluster 4 secretes colony stimulating factor 1 (*Csf1*) which also shows potential regulation of cluster 3 through *Csf1r*. Overall, the trajectory and network analyses support potential cell plasticity and lineage relationships and cell–cell communications across OEC subclusters.

## Spatial confirmation of defined OEC subclusters within the ONL in situ

Here we confirm, at the protein level, that some of the top genes derived from cultured OECs clusters are expressed in sections of the olfactory system from 8- to 10-week-old female Sprague-Dawley rats. Our results illustrate the spatial distribution of two to three top genes from clusters 0 to 3 in the ONL (*Figure 7*; corresponding tSNE plots in *Figure 7—figure supplement 1* for comparison). In cluster 0 we show high levels of Peripheral myelin protein 22 (Pmp22) in OECs (*Figure 7a*) with the remainder of the olfactory bulb unlabeled. The small protein Gliomedin (Gldn) is a component of Schwann cell microvilli and facilitates the formation of peripheral nodes of Ranvier (*Eshed et al., 2005*). Here we detect numerous small dot-like structures that overlay the ONL (*Figure 7b, c*). The detection of high levels of Gldn in nonmyelinating OECs without nodes of Ranvier is surprising and suggests that it may also function as a glial ligand associated with OSN axons. For cluster 1, we examined the expression of the important axonal growth factor Activating transcription factor 3 (Atf3) and confirmed that OECs express it widely (*Figure 7d*). Growth associated protein 43 (Gap43), a well-established marker of OECs and immature OSNs, is highly expressed in the ONL and at a lower level in the glomeruli of the olfactory bulb (*Figure 7e*). Nestin (Nes), an intermediate filament associated with neural stem cells, is widely expressed in the ONL (*Figure 7f*).

Compared to the two large clusters, the remaining clusters contain more specialized OECs. The top marker in the proliferative cluster 2 is Stathmin1 (Stmn1), encoding a microtubule destabilizing protein found in areas of adult neurogenesis (*Boekhoorn et al., 2014*). Stathmin 1 immunoreactivity (*Figure 7g*, *Figure 7—figure supplement 1*) is detected in all OEC clusters and immature OSNs, and therefore fills the ONL in a pattern similar to that seen in Gap43 (*Figure 7e*). Two markers associated with the cell cycle, Ube2c (*Figure 7h*) and Cdk1 (not shown), identify cluster 2 cells in the ONL, as well as the early progenitor marker Mki67 (*Figure 7i*). Cluster 3 is strongly associated with OEC immune function and the expression of Apolipoprotein E (Apoe), Annexin A3 (Anxa3), and Allograft inflammatory factor 1 (Aif1/Iba1) were confirmed. Apoe (*Figure 7j*) is detected throughout the ONL and includes some cellular structures. Expression of Anxa3 (*Figure 7k*) and Aif1/Iba1 (*Figure 7l*) is similar – distinct small cells labeled in the ONL. Our smallest group, cluster 4, was difficult to confirm with selected antibodies targeting the top marker genes, likely due to the limited number of cells in this cluster. Overall, our in situ experiments confirmed the presence and distribution of four out of the five OEC subtypes detected by scRNA-seq.

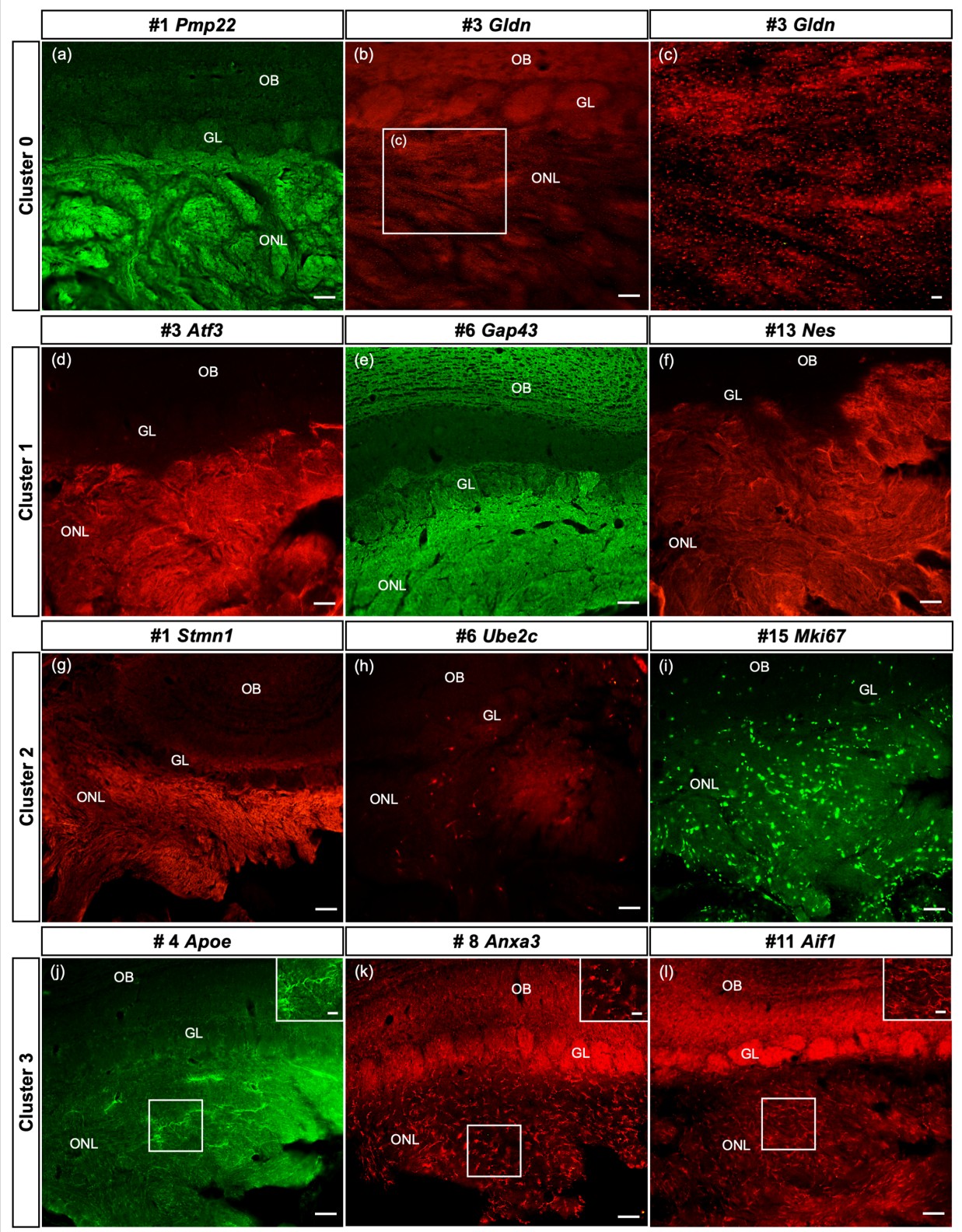

**Figure 7.** Spatial confirmation of the defined olfactory ensheathing cells (OECs) subclusters within the olfactory nerve layer (ONL). The protein expression of a number of top 20 genes from this single-cell RNA-sequencing (scRNA-seq) study of purified OEC cultures is verified in olfactory bulb sections. In all images, the ONL (layer I) is at the bottom of the image, the glomeruli (GL, layer II) next, and the remainder of the olfactory bulb toward the top. *Figure 7—figure supplement 1* illustrates the corresponding t-distributed stochastic neighbor embedding (tSNE) plots for each gene in the

*Figure 7 continued on next page*

Figure 7 continued

figure. (**a–c**) The top gene in the largest cluster, Peripheral myelin protein 22, is highly expressed throughout the ONL. Gliomedin, the third-ranked gene in cluster 0, is detected as small discrete dot-like structures overlaying the ONL (b, box enlarged in c). (**d–f**) OECs in cluster 1 are immunolabeled by antibodies against the axonal growth factor Atf3 (**d**) and the intermediate filament Nestin (**f**). High levels of Gap43 in the ONL (**e**) are due to expression by OECs and axons of immature olfactory sensory neurons. (**g–i**) Strong immunoreactivity of Stathmin-1 in the ONL reflects labeled axons from immature olfactory sensory neurons and OECs. Ube2c, a G2/M cell cycle regulator, is expressed in a small number of cells in the ONL, whereas Mki67-labeled cells are widespread. (**j–l**) The immune function of this cluster is confirmed by antibodies against Apoe (**j**), Anxa3 (**k**), and Aif1/Iba1 (**l**), markers expressed by microglia and macrophages. Scale bars: a, b, d–l = 50 µm; c, insets in j–l = 25 µm.

The online version of this article includes the following figure supplement(s) for figure 7:

**Figure supplement 1.** The t-distributed stochastic neighbor embedding (tSNE) plots corresponding to the OEC subclusters are illustrated.

## OECs synthesize and secrete Reelin

Reelin (*Reln*) is detected in OECs in this study (*Figures 2f and 8a, g*) and reported in previous microarray studies (*Guérout et al., 2010*; *Roet et al., 2011*). However, the presence of the *Reln* gene within OECs is disputed in the literature (*Dairaghi et al., 2018*; *Schnaufer et al., 2009*). *Reln* codes for a large ECM glycoprotein that is secreted by neurons (*D'Arcangelo et al., 1995*; *Kubasak et al., 2004*), but also by Schwann cells (*Panteri et al., 2006*). In the canonical Reelin-signaling pathway, Reelin binds to the very-low-density lipoprotein receptor (Vldlr) and apolipoprotein E receptor 2 (ApoER2) and induces Src-mediated tyrosine phosphorylation of the intracellular adaptor protein Disabled-1 (Dab1). Both Reelin and Dab1 are highly expressed in embryos and contribute to correct neuronal positioning. A recent study claimed that peripheral OECs express high levels of Dab1 and Vldlr (*Dairaghi et al., 2018*), and that Reelin was expressed only by mesenchymal cells located near the cribriform plate, not in OECs (*Dairaghi et al., 2018*; *Schnaufer et al., 2009*). Our scRNA-seq findings of OECs demonstrated low expression levels of *Dab1* and *Apoer2* (*Figure 8b, c*), moderate expression of *Vldlr* (*Figure 8d*) and substantial levels of *Reln* in OEC subclusters 0–3 (*Figure 8a*). Interestingly, OECs also contain proteolytic enzymes that cleave the 400Kd secreted Reelin. Tissue plasminogen activator (tPA; *Figure 8e*) cleaves Reln at its C-terminus, and the matrix metalloproteinase ADAMTS-4 cuts at both the N- and C-cleavage sites of Reln (*Figure 8f*; *Krstic et al., 2012*).

The confusion about Reelin expression in OECs likely stemmed from the use of standard immunochemical methods to confirm its presence. Our attempts to detect Reelin expression in the adult ONL or in cultured OECs were unsuccessful using several antibodies and multiple protocols, despite consistent localization of Reelin within mitral cells of the olfactory bulb. To understand this discrepancy, we examined Reelin expression in embryonic sections which included both the peripheral and OB-OECs. Images of E16.5 *Reln*[+/+] olfactory system show Reelin expression in peripheral OECs and in neurons of the olfactory bulb (*Figure 8g*, green), but not on OB-OECs in the ONL or in sections of *Reln*[−/−] mice (*Figure 8g, g2, h*). In comparison, Blbp expression is uniformly expressed in the ONL of both peripheral and OB-OECs (*Figure 8g1, h1*, red). Enlargements of the region where axon bundles from OSNs join the ONL show that Blbp is present in OB-OECs throughout the ONL, but Reelin is detected only on peripheral OECs (*Figure 8g, g1, g2*).

To determine if adult rat OB-OECs synthesize and secrete Reelin in vitro, we treated primary Ngfr[p75]-purified OECs with or without Brefeldin-A, a fungal metabolite that specifically inhibits protein transport from the endoplasmic reticulum to the Golgi apparatus (*Misumi et al., 1986*). In Brefeldin-treated primary OEC cultures, spindle-shaped GFP-labeled OECs contain Reelin in their cytoplasm (*Figure 9a*, arrows). Large contaminating Reln-immunonegative cells present in primary cultures serve as an intrinsic control (*Figure 9a*, arrowhead). Reelin expression was also confirmed in Brefeldin-treated Ngfr[p75]-purified OEC cultures (*Figure 9b, c*), a finding which supports our scRNA-seq results.

To further confirm that cultured OECs express and secrete Reelin, we performed western blotting. The G10 antibody recognizes full-length Reelin protein, which is approximately 400 kDa, and its two cleaved fragments at ~300 and ~180 kDa (*Lambert de Rouvroit et al., 1999*). Both the rat ONL and *Reln*[+/+] mouse olfactory bulbs showed the characteristic large molecular weight band at 400 kDa and another at 150 kDa (*Figure 9d*, lanes 1 and 2, Reln). As expected, these bands were absent in mutant *Reln*[−/−] olfactory bulbs (*Figure 9d*, lane 3). Additionally, Reelin-positive bands were present in blots of OEC whole-cell lysates (WCL), representing Reelin synthesized in OECs, and OEC conditioned media (CM; *Figure 9d*, lanes 4–8 Reln), representing Reelin secreted by OECs. A 4–15% gradient gel allowed for the visualization of all three Reelin isoforms (*Figure 9e*). Brain extracts of *Reln*[+/+] (positive control)

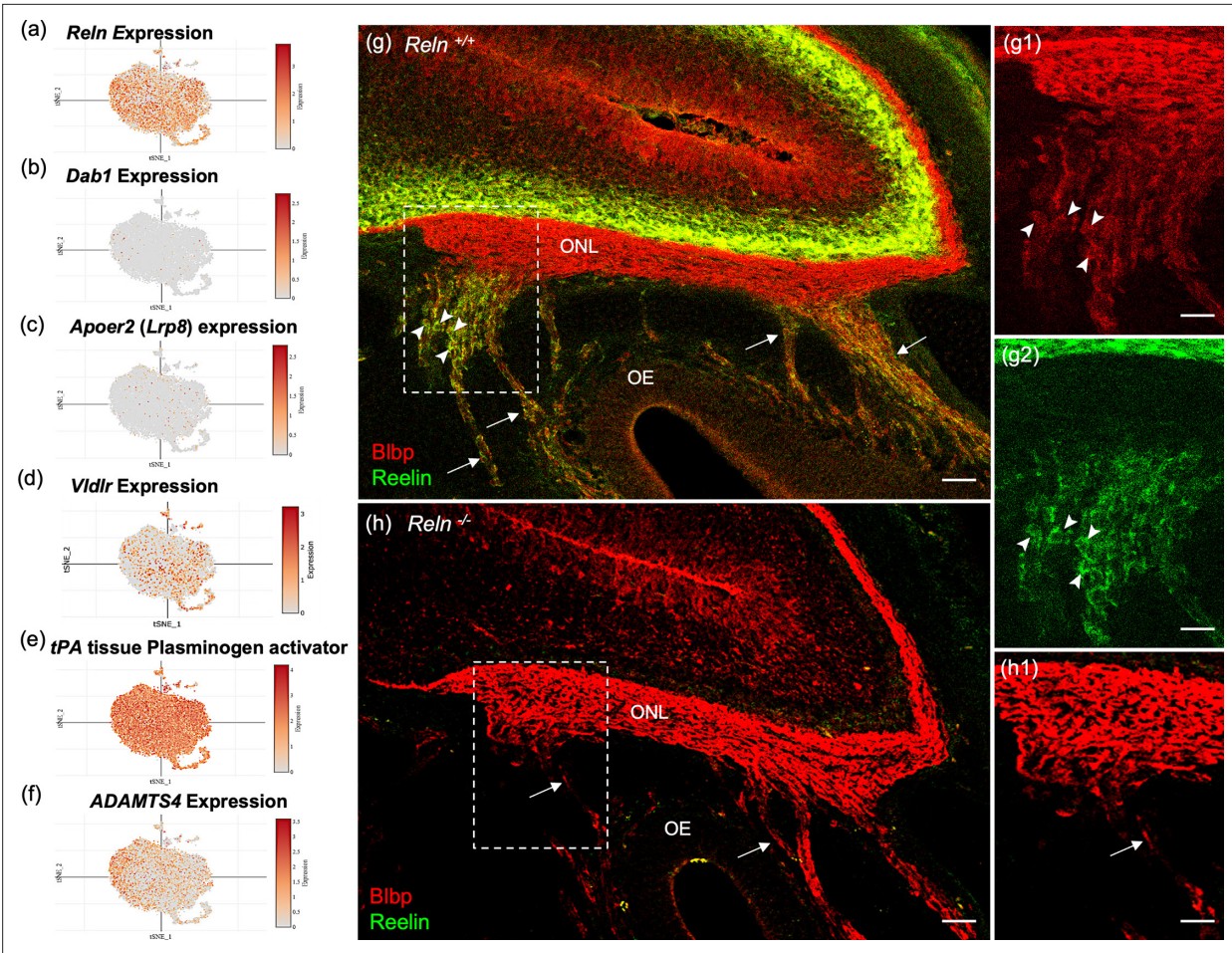

**Figure 8.** Reelin-signaling pathway marker genes in olfactory ensheathing cell (OEC) clusters and Reelin immunoreactivity in embryonic olfactory system. (**a–f**) These six t-distributed stochastic neighbor embedding (tSNE) plots show OEC gene expression levels associated with Reelin signaling. *Reln* is highly expressed in most OEC clusters (**a**), *Disabled-1* (*Dab1*, **b**), and *Apolipoprotein E receptor 2* (*Apoer2*, **c**) show little expression, and *Very-low-density lipoprotein receptor* has moderate expression (*Vldlr*, **d**). The serine protease tissue Plasminogen Activator (e, *tPA*) is highly expressed in all OEC subsets and cleaves secreted Reelin only at its C-terminus. ADAMTS-4 cleaves both the N- and C-terminals of Reelin, and is expressed at low levels by OECs (**f**). (**g–g2**) Sagittal section of the olfactory epithelium (OE) and olfactory bulb from an E16.5 *Reln*⁺/⁺ mouse immunolabeled for Blbp (**g**, **g1**; red) and Reelin (**g**, **g2**; green). Boxed area in g is enlarged in g1 and g2. Axons of olfactory sensory neurons (arrows) are surrounded by peripheral OECs that express both Blbp (g1, arrowheads) and Reelin (**g2**, arrowheads). High Reln expression is obvious in olfactory bulb neurons (**g**, **g1**, green) but the olfactory nerve layer (ONL) only expresses Blbp (**g**, **g2**). (**h**, **h1**) No Reelin is detected in a sagittal section from a *Reln*⁻/⁻ mouse, yet Blbp expression in the ONL appears normal. Scale bars: g–h = 50 µm; g1, g2, h1 = 50 µm.

and *Reln*⁻/⁻ (negative control) mice confirm the specificity of Reelin detection (*Figure 9e*, lanes 1 and 2, *Figure 9f*). All three Reelin isoforms at their corresponding molecular weights are present in rat ONL and OEC CM samples (*Figure 9e*, lanes 3–6, *Figure 9f*). Quantification of the band densities from 4% to 15% gels showed that cultured OECs derived from the rat ONL and CM produce significantly higher levels of 400 and 300 kDa Reelin compared to *Reln*⁻/⁻ mice (*Figure 9f*, p < 0.05).

## Discussion

Our scRNA-seq study of immunopurified OECs identified five OEC subtypes with distinct gene expression patterns, pathways, and properties of each subtype. The well-established OEC markers are widely distributed throughout most clusters derived from cultured OECs. We also experimentally confirmed a number of the top markers at the protein level in the OEC subtypes on sections of the rat olfactory system and established that OECs secrete both Reelin and Ctgf, ECM molecules reported to be important for neural repair and axon outgrowth. Additional proregenerative OEC marker genes

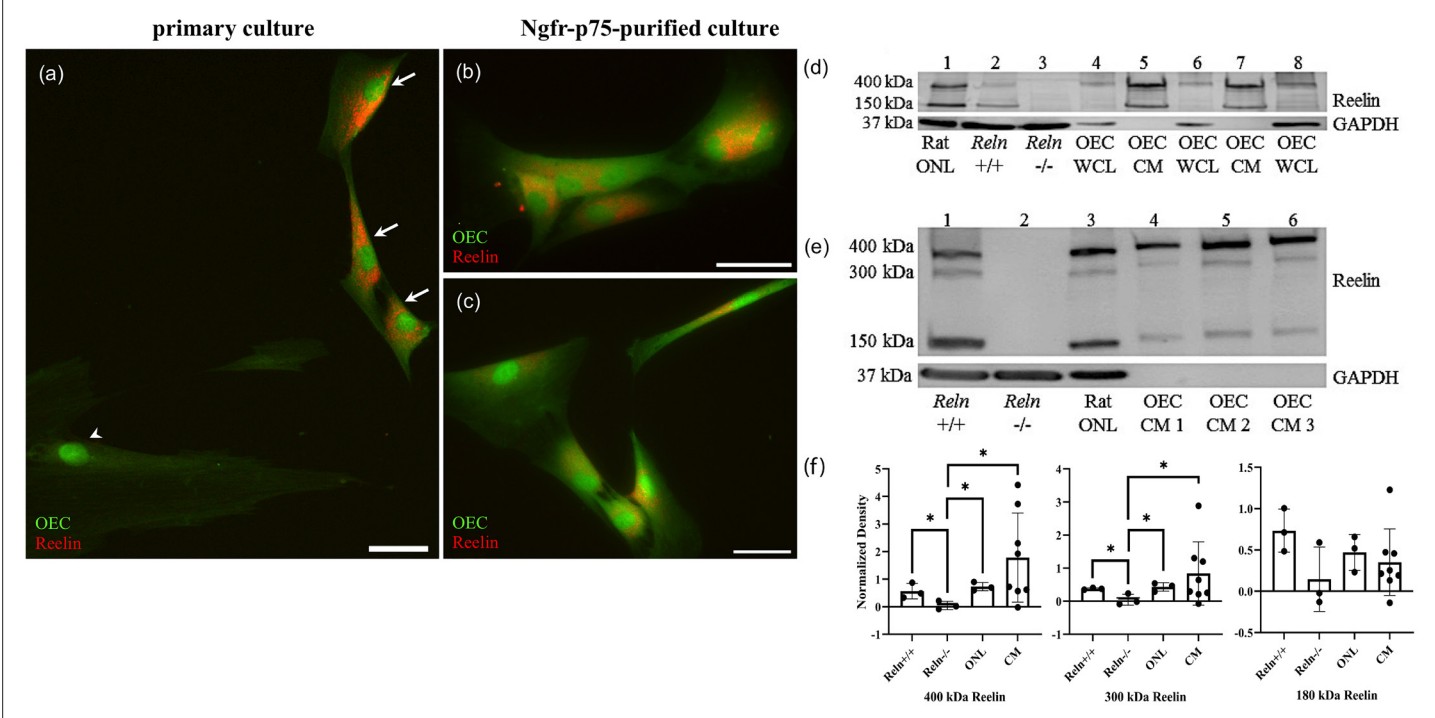

**Figure 9.** Reelin is expressed and secreted by olfactory ensheathing cells (OECs). (**a–c**) Rat OEC cultures were treated with Brefeldin-A to inhibit protein transport and subsequent secretion. Reelin immunoreactivity (red) in a primary culture was detected in green fluorescent protein (GFP)-labeled OECs (a, arrows), but not in another cell type (arrowhead). GFP-labeled OECs which were immunopurified with anti-Ngfr$^{p75}$ also express Reelin (red; **b, c**). (**d**) Western blot confirms the expression of Reelin in rat Olfactory Nerve Layer I and II (ONL; lane 1 of western blot). $Reln^{+/+}$ and $Reln^{-/-}$ mouse olfactory bulbs were used as positive and negative controls, respectively (lanes: 2 and 3). Reelin synthesized by cultured OECs was found at low levels in whole-cell lysates (WCL; lanes: 4, 6, and 8), compared to Reelin secreted by cultured OECs into tissue culture medium, termed OEC 'conditioned medium' (CM; lanes: 5 and 7). GAPDH was the loading control for tissue homogenates (lanes: 1–4, 6, and 8). (**e**) All three Reelin isoforms (400, 300, and 150 kDa) were detected with a 4–15% gradient gel. $Reln^{+/+}$ and $Reln^{-/-}$ mouse cortices were used as controls (lanes: 1 and 2). Reelin was detected in the rat ONL (lane 3) and in three rat OEC-CM samples (lanes: 4, 5, and 6). GAPDH was the loading control for tissue homogenates (lanes: 1–3). (**f**) Quantitation of multiple isoforms of Reelin from 4% to 15% gradient gels. Positive and negative controls are $Reln^{+/+}$ and $Reln^{-/-}$ mouse cortices. Both rat tissue from the ONL (n = 3) and CM (n = 9) contain significantly more 400 and 300 kDa Reelin than the $Reln^{-/-}$ mice. Bars represent the standard deviation of the mean. One-sided Mann–Whitney $U$ test was used to test that protein expression levels in the other groups are greater than those in the $Reln^{-/-}$ group, indicative of significant expression of $Reln$ in the test groups. *p < 0.05. Scale bars: a–c = 40 µm.

The online version of this article includes the following source data for figure 9:

**Source data 1.** Original images for the western blots shown in *Figure 9d and e*.

**Source data 2.** Labeled images for the western blots shown in *Figure 9d and e*.

such as *Atf3*, *Btc*, and *Pcsk1* also were identified. Our findings provide an unbiased and in-depth view of the heterogeneity of OEC populations with demonstrated injury repair capacity, and the potential mechanisms that underlie their regenerative and protective properties.

This scRNA-seq analysis confirmed the immunopurification of the OECs compared with the mixture of OECs grown with numerous fibroblasts and microglia. Although some may debate if purified OECs are needed to maximize neural repair, this study suggests that immunopurified OECs have higher expression of anti-stress genes than unpurified controls. Several of our earlier studies, which transplanted similarly purified OECs, also showed evidence of neural regeneration following severe SCI (*Khankan et al., 2016*; *Takeoka et al., 2011*; *Thornton et al., 2018*). In addition to the purity of OECs, we found significant overlap of the marker genes between our study and those from the meta-analysis of five OEC microarray studies on cultured OB-OECs (*Roet et al., 2011*), supporting the reproducibility of our scRNA-seq findings. When we compared our data from cultured OB-OECs to previous scRNA-seq studies of the olfactory system tissue which included peripheral OECs (*Durante et al., 2020*), or a mixture of peripheral and OB-OECs (*Tepe et al., 2018*), there was little overlap. Most likely, the substantial differences were due to the dissection areas, techniques used for OEC

preparation, the species studied, as well as sex and age of the subjects. It is important to note that our OB-OEC populations were repeatedly shown to induce injury repair, and their subtypes and markers support their diverse reparative functions, whereas the neural repair function of the cell populations in the other scRNA-seq studies using different tissue preparations is unclear.

## OECs are hybrid glia that promote neural repair

Based on intrinsic gene expression patterns revealed by scRNA-seq data, there is expression of select markers for Schwann cells, astrocytes, oligodendrocytes and microglia within our five clusters of OECs. We observed transcriptionally distinct clusters that highly expressed select markers for proliferation (cluster 2), microglia (cluster 3), and astrocytes and oligodendrocytes (cluster 4). The microglia/macrophage-like cells reported by *Smithson and Kawaja, 2010* do resemble our transcriptome-based subtyping of OEC cluster 3. Our results further support the activities of OECs in modulating immune responses, based on the expression of genes involved in the endocytosis of bacteria, the regulation of microglia activation, and the expression of numerous lysosomal pathways (*Leung et al., 2008*). OECs in cluster 4 are distinctive by their expression of several astrocytic markers that regulate synaptogenesis (Thrombospondin 2) and synaptic connectivity, and remove inactive synaptic connections (Complement component 1q family; *Christopherson et al., 2005*; *Eroglu and Barres, 2010*). Additionally, our trajectory and network analyses indicate potential relationships between the OEC subclusters, which warrant further experimental testing.

Both OECs and Schwann cells originate from the neural crest, express similar immunological markers, and share many transcriptional similarities (*Barraud et al., 2010*; *Forni et al., 2011*; *Vincent et al., 2005*). OECs share phenotypic characteristics with myelin-producing Schwann cells and oligodendrocytes, but more closely resemble Schwann cells (*Doucette, 1991*; *Radtke et al., 2011*; *Ramón-Cueto and Valverde, 1995*). Gliomedin is a required ECM protein that initiates the assembly of nodes of Ranvier along Schwann cells (*Eshed et al., 2005*), yet despite the fact that OECs do not have nodes of Ranvier, they express high levels of Gliomedin. This novel Gliomedin expression in OECs suggests a possible function as a glial ligand for the unmyelinated axon bundles that OECs ensheath. Following peripheral nerve injury, the loss of contact between axons and myelinating Schwann cells induces a nonmyelinating phenotypic change in Schwann cells (*Fu and Gordon, 1997*). These dedifferentiated nonmyelinating Schwann cells are proregenerative and upregulate the expression of Glial fibrillary acidic protein (Gfap; *Jessen et al., 1990*), Ngfr[p75] receptor (*You et al., 1997*), cell adhesion molecules L1 and N-Cam (*Martini and Schachner, 1988*), and neurotrophic factors, including Bdnf (*Meyer et al., 1992*) and Glial cell-line derived neurotrophic factor (*Trupp et al., 1995*). Genes for these proregenerative factors are detected in OECs in the present study, as well as the transcription factor *Mitf*, that controls a network of genes related to plasticity and repair in Schwann cells (*Daboussi et al., 2023*).

Our in vitro and in vivo experimental validation of numerous marker genes at the protein level strongly support their validity. Based on our tissue localization, the two large clusters (0 and 1) were found throughout the ONL, whereas cluster 2 (progenitors) and cluster 3 (microglial) are scattered within the ONL. Future functional studies of these subclusters will benefit from both the marker genes and their localizations uncovered in this study.

## Reelin and CTGF are distinctive markers for OECs

Reelin is a developmentally expressed protein detected in specific neurons, in addition to OECs and Schwann cells. The canonical Reelin-signaling pathway involves neuronal-secreted Reelin binding to Vldlr and ApoER2 expressed on Dab1-labeled neurons. Following Reelin binding, Dab1 is phosphorylated by Src family kinases which initiates multiple downstream pathways. Very little is known, however, about Reelin secreted by glia. *Panteri et al., 2006* reported that Schwann cells express low levels of Reelin in adults, and that it is upregulated following a peripheral nerve crush, as is reported above for many neurotrophic factors. Reelin loss in Schwann cells reduced the diameter of small myelinated axons but did not affect unmyelinated axons (*Panteri et al., 2006*). In the olfactory system, OECs ensheath the Dab1-labeled, unmyelinated axons of OSNs which are continuously generated and die throughout life. OEC transplantation following SCI would provide an exogenous source of Reelin that could phosphorylate Dab1-containing neurons or their axons. Dab1 is expressed at high levels in the axons of some projection neurons, such as the corticospinal pathway (*Abadesco et al.,*

*2014*). Future experiments are needed to explore the function that glial-secreted Reelin may have on axonal regeneration.

Adult mammals show little evidence of spontaneous axonal regeneration after a severe SCI in contrast to transected neonatal rats (*Bregman, 1987*; *Bregman et al., 1993*) and young postnatal opossums (*Lane et al., 2007*). In immature mammals, axons continue to project across or bridge the spinal cord transection site during development. Lower organisms such as fish, show even more evidence of regeneration following severe SCI. *Mokalled et al., 2016* reported that glial secretion of Ctgfα/Ccn2 was both necessary and sufficient to stimulate a glial bridge for axon regeneration across the zebrafish transection site. Cells in the injury site that express Ctgf include ependymal cells, endothelial cells, and reactive astrocytes (*Conrad et al., 2005*; *Mokalled et al., 2016*; *Schwab et al., 2001*). Here we show that, although rare, Ctgf-positive OECs can contribute to glial bridge formation in adult rats. The most consistent finding among our severe SCI studies combined with OEC transplantation is the extent of remodeling of the injury site and axons growing into the inhibitory lesion site, together with OECs and astrocytes. The formation of a glial bridge across the injury was critical to the spontaneous axon generation seen in zebrafish (*Mokalled et al., 2016*) and likely contributed to the axon regeneration detected in our OEC transplanted, transected rats (*Dixie, 2019*; *Khankan et al., 2016*; *Takeoka et al., 2011*; *Thornton et al., 2018*).

## Limitations in these OEC scRNA-seq studies

We recognize that this study is a single snapshot of OEC gene expression derived from adult female rats before they are transplanted above and below the spinal cord transection site. We would expect the gene expression of transplanted OECs to change in each new environment, that is as they migrate into the injury site, integrate into the glial scar, and wrap around axons. Based on our past studies, OECs survived in an outbred Sprague-Dawley female rat model for ~4 weeks (*Khankan et al., 2016*) and in an inbred Fischer 344 female model for 5–6 months (*Dixie, 2019*). As SCI transplant procedures are further enhanced and OEC survival improves, these hybrid glial cells should be examined at multiple time points after transplantation to better evaluate their proregenerative characteristics.

We acknowledge that in humans, males account for ~80% of spinal cord injuries (*National Spinal Cord Statistical Center, 2024*) and sustain more serious urinary tract issues than females. We examined females in the current study due to practical experimental considerations, but it is necessary to examine males in future studies. We also did not modulate any of the genes or proteins in the identified OEC subtypes to test their causal and functional roles, thus our findings remain correlative. Future gene/protein modulation studies are necessary to understand the functional roles of the individual OEC subtypes in the context of their reparative functions to determine which pathways and subtypes are more critical and can be enhanced for neural repair. Our current findings build the foundation for these future studies to help resolve the role of OECs in SCI repair.

Extensive differences between OEC preparations contribute to the large variation in results from OEC treatments following SCI. This scRNA-seq study focused entirely on OB-OECs, and the next step would be to carry out similar studies on the peripheral, lamina-propria-derived OECs to discern the differences between the two OEC populations. Such comparative studies using scRNA-seq will help define the underlying mechanisms and resolve the variability in results from OEC-based therapy. Detailed studies of the composition of different OEC transplant types will contribute to identifying the most reparative cell transplantation treatments.

## Conclusion

To understand how immunopurified OECs repair spinal cord injuries, we carried out an unbiased scRNA-seq analysis and found that OECs have a wide range of subtypes including a proliferative state, microglia-like cells and neural regenerative clusters. We demonstrated that these multi-functional OECs are clearly distinguishable from each other by expression of unique genes and pathways within subtypes based on the scRNA-seq data and our in vivo tissue studies. These subtypes of OECs produce important ECM genes, including *Reln* and *Ctgf* and many other signaling molecules which

are involved in a wide range of intercellular communication as well as axon regeneration. Our single-cell resolution investigation offers a comprehensive molecular landscape of OECs and numerous novel targets for future injury repair and functional/mechanistic studies.

# Materials and methods

**Key resources table**

| Reagent type (species) or resource | Designation | Source or reference | Identifiers | Additional information |
|---|---|---|---|---|
| Strain, strain background (mouse) | Reeler mouse (B6C3Fe-ala-Reln^rl) | Jackson Laboratory | Strain 000235 | Used for OEC mouse tissue sections cultures, and western blots |
| Strain, strain background (rat) | GFP-expressing Sprague-Dawley rats (SD-Tg) | SLC, Shizuoka, Japan7 | Z-004 SD-Tg (Act-EGFP) | Used for breeding |
| Strain, strain background (rat) | Unlabeled Sprague-Dawley female rats (SD) | Charles Rivers Laboratory | Strain 001 CD IGS | Used for breeding with SD-Tg males |
| Strain, strain background (rat) | GFP-SD-Tg males bred with SD females | Sources noted above | Used to derive 8- to 10-week-old female GFP-SD-Tg rats | OECs used for scRNA-Seq analysis and rat cultures |
| Antibody | Atf3; Mouse monoclonal | Abcam | Cat# ab191513, RRID:AB_2868437 | 1:4000; TSA + citric acid |
| Antibody | Aif1/Iba1, Rabbit polyclonal | Wako (Osaka, Japan) | Cat# 019-19741, RRID:AB_839504 | 1:5000; IF |
| Antibody | Anxa3, Rabbit polyclonal | Gene Tex | Cat# GTX103330, RRID:AB_11174712 | 1:5000; TSA + citric acid |
| Antibody | Apolipoprotein E (Apoe), Goat polyclonal | Sigma-Aldrich | Cat# AB947, RRID:AB_2258475 | 1:2000; TSA + citric acid |
| Antibody | Brain lipid-binding protein (Blbp), Rabbit monoclonal | Sigma-Aldrich | Cat# ZRB13190, RRID:AB_2920766 | 1:5000; IF |
| Antibody | Brain lipid-binding protein (Blbp), Rabbit polyclonal | Sigma-Aldrich | Cat# ABN14, RRID:AB_10000325 | 1:1500; IF |
| Antibody | Connective tissue growth factor (Ctgf, Ccn2), Rabbit polyclonal | Abcam | Cat# ab6992, RRID:AB_305688 | 1:500, IF |
| Antibody | Gliomedin (Gldn), Rabbit polyclonal | Abcam | Cat# ab24483, RRID:AB_2111616 | 1:100; IF + citric acid |
| Antibody | Growth-associated protein (Gap43), Rabbit polyclonal | Abcam | Cat# ab16053, RRID:AB_443303 | 1:500; IF |
| Antibody | Green fluorescent protein (GFP), Chicken polyclonal | Aves Labs Inc | Cat# GFP-1020, RRID:AB_10000240 | 1:1000; IF |
| Antibody | Glial fibrillary acidic protein (GFAP), Mouse monoclonal | BD Biosciences | Cat# 556327, RRID:AB_396365 | 1000; IF |
| Antibody | Glial fibrillary acidic protein (GFAP), goat polyclonal | Sigma | Cat# SAB2500462, RRID:AB_10603437 | 1:2000; IF |
| Antibody | Glyceraldehyde-3-phosphate dehydrogenase (GAPDH), Mouse monoclonal | Sigma-Aldrich | Cat# MAB374, RRID:AB_2107445 | 1:100K; WB |
| Antibody | L1, Rabbit polyclonal | Gift from Dr. V Lemmon **Brittis et al., 1995** | | 1:10,000; IF |
| Antibody | Marker of proliferation ki-67 (Mki67), Rabbit polyclonal | Abcam | Cat# ab15580, RRID:AB_443209 | 1:500; IF 1:5000; TSA |
| Antibody | Nestin (Nes), Mouse monoclonal | Developmental Studies Hybridoma Bank | Cat# Rat-401, RRID:AB_2235915 | 1:1000; TSA + citric acid |
| Antibody | Neurofilament-200 (NF), Rabbit polyclonal | Sigma-Aldrich | Cat# AB1989, RRID:AB_91202 | 1000; IF |

*Continued on next page*

*Continued*

| Reagent type (species) or resource | Designation | Source or reference | Identifiers | Additional information |
|---|---|---|---|---|
| Antibody | N-Cadherins, Mouse monoclonal | Invitrogen | Cat# 33-3900, RRID:AB_2313779 | 1:500; IF |
| Antibody | Neural cell adhesion molecule (NCAM), Rabbit polyclonal | Sigma | Cat# AB5032, RRID:AB_2291692 | 1:5000; IF |
| Antibody | Nerve growth factor receptor, p75 (Ngfr$^{p75}$), Mouse monoclonal | Hybridoma gift from Dr. EM Shooter (*Chandler et al., 1984*) | 192-IgG | 1:50; IF 1:5; immuno-purification for OECs |
| Antibody | Nerve growth factor receptor, p75 (Ngfr$^{p75}$), Rabbit polyclonal | Sigma-Aldrich | Cat# AB1554, RRID:AB_90760 | 1:30K; IF 1:1500; immuno-purification |
| Antibody | Peripheral myelin protein 22 (Pmp22), Rabbit polyclonal | Gene Tex | Cat# GTX85834, RRID:AB_10733560 | 1:1000; TSA + MeOH* |
| Antibody | Reelin (Reln), Goat polyclonal | R&D Systems | Cat# AF3820, RRID:AB_2253745 | 1:250; IF |
| Antibody | Reelin (Reln), Mouse monoclonal G10 | Sigma-Aldrich | Cat# MAB5364, RRID:AB_2179313 | 1:1000; F 1:750; WB |
| Antibody | S100 calcium-binding protein β (S100β), Rabbit polyclonal | Dako A/S (Glostrup, Denmark) | Cat# GA50461-2, RRID:AB_2811056 | 1:500; IF |
| Antibody | Serotonin (5HT), Goat polyclonal | Immunostar | Cat# 20079, RRID:AB_572262 | 5000; IF |
| Antibody | Stathmin 1 (Stmn1), Rabbit monoclonal | Abcam | Cat# ab52630, RRID:AB_2197257 | 1:7000; TSA + citric acid |
| Antibody | SRY-Box transcription factor 10 (SOX10), Goat polyclonal | R&D Systems | Cat# AF2864, RRID:AB_442208 | 1:75; ICC |
| Antibody | Ubiquitin-conjugating enzyme E2 C (Ube2c), Rabbit monoclonal | Abcam | Cat# ab252940, RRID:AB_2910263 | 1:1000; TSA + citric acid |
| Antibody | Alexa Fluor 488 AffiniPure donkey anti-Chicken IgY | Jackson Immuno Research | Cat# 703-545-155, RRID:AB_2340375 | 1:200–500 |
| Antibody | Alexa Fluor 488 AffiniPure donkey anti-goat | Jackson Immuno Research | Cat# 705-545-003, RRID:AB_2340428 | 1:250 |
| Antibody | Alexa Fluor 488 AffiniPure donkey anti-mouse | Jackson Immuno Research | Cat# 715-545-150, RRID:AB_2340846 | 1:500 |
| Antibody | Alexa Fluor 488 donkey anti-rabbit | Invitrogen | Cat# A21206, RRID:AB_2535792 | 1:250 |
| Antibody | Alexa Fluor 555 donkey anti-mouse | Life Technologies | Cat# A31570 RRID:AB_2536180 | 1:200–500 |
| Antibody | Alexa Fluor 555 donkey anti-rabbit | Life Technologies | Cat# A31572, RRID:AB_162543 | 1:500 |
| Antibody | Alexa Fluor 647 donkey anti-rabbit | Jackson Immuno Research | Cat# 711-605-152, RRID:AB_2492288 | 1:200–500 |
| Antibody | Biotin-SP AffiniPure donkey anti-mouse IgG | Jackson Immuno Research | Cat# 715-065-150, RRID:AB_2307438 | 1:1000 |
| Antibody | Biotin-SP AffiniPure donkey anti-rabbit IgG | Jackson Immuno Research | Cat# 711-065-152, RRID:AB_2340593 | 1:1000 |
| Antibody | Biotin-SP AffiniPure goat anti-rabbit IgG, FC fragment specific | Jackson Immuno Research | Cat#111-065-008, RRID:AB_2337959 | 1:1000 hydrophobic Petri dishes |
| Antibody | Biotinylated horse anti-goat | Vector Laboratories | Cat# BA-9500, RRID:AB_2336123 | 1:1000 |
| Other | Streptavidin-conjugated horseradish peroxidase | PerkinElmer | Cat# NEL750001ET, RRID:AB_2617185 | 1:1000 |

*Continued*

| Reagent type (species) or resource | Designation | Source or reference | Identifiers | Additional information |
|---|---|---|---|---|
| Commercial assay or kit | Tyramide Signal Amplification (TSA) plus Fluorescein | PerkinElmer | Cat# NEL741001KT | 1:150 |
| Commercial assay or kit | Tyramide Signal Amplification (TSA) plus Cyanine 3 | PerkinElmer | Cat# NEL74400KT | 1:100 |

Sections underwent a permeabilization step with 100% methanol for 20 min at −20°C.
Abbreviations: IF, immunofluorescence; WB, western blot.

## Animals

All of the experimental procedures used on animals were approved by the Chancellor's Animal Research Committee at UCLA and carried out in accordance with the National Institutes of Health guidelines. Animals were housed in the Terasaki Life Science Building vivarium under standard conditions with free access to food and water.

### Sprague-Dawley rats

We bred our male GFP-expressing Sprague-Dawley rats (*Perry et al., 1999*) with unlabeled female Sprague-Dawley rats (Charles Rivers Laboratory), which were used previously to obtain GFP-labeled OECs from female rats for three of our SCI studies (*Dixie, 2019*; *Khankan et al., 2016*; *Thornton et al., 2018*) in order to unambiguously identify the transplanted GFP-positive OECs near the injury sites. A total of 8, 8- to 10-week-old GFP-labeled female rats were used to obtain GFP-OECs for scRNA-seq. Only females were used in order to match the sex of previous SCI studies conducted exclusively on female rats (*Dixie, 2019*; *Khankan et al., 2016*; *Takeoka et al., 2011*; *Thornton et al., 2018*). Following complete thoracic spinal cord transection, an adult rat is unable to urinate voluntarily and therefore urine must be manually 'expressed' twice a day throughout the experiment. Females have a shorter urethra than males, and thus their bladders are easier to empty completely. An additional 4, 8- to 10-week GFP-labeled rats were used to isolate Reelin protein for western blotting and 6, 8- to 10-week unlabeled female Sprague-Dawley rats were used to confirm gene expression in the five OEC subtypes. An overdose of ketamine–xylazine or pentobarbital was used for euthanasia before either the rat olfactory bulbs were removed or the rats were perfused with 4% paraformaldehyde or 2% PLP (2% paraformaldehyde; 0.075 M lysine-HCL-monohydride; 0.010 M sodium periodate; 0.1 M sodium phosphate) followed by a 1- to 2-hr postfix.

### *Reeler* mice

To study Reelin expression in OECs, heterozygous ($Reln^{+/-}$) mice, originally purchased from the Jackson Laboratory (B6C3Fe-*ala-Reln*$^{rl}$, Bar Harbor, ME), were used for breeding. Genotypes of these mice were identified by PCR (*D'Arcangelo et al., 1996*). To obtain timed pregnancies, $Reln^{+/-}$ mice were bred and checked every morning for plugs. Pregnant mice were heavily anesthetized with pentobarbital (100 mg/kg) and embryos were extracted at embryonic day 16.5. Heads were immersed in 2% PLP, washed in PB, cryoprotected, and embedded in Shandon M1 embedding matrix (Thermo Scientific, 1310).

## Preparation of OB-OEC cultures from rats and mice

Olfactory bulbs were collected from 8- to 10-week-old GFP-labeled Sprague-Dawley rats or $Reln^{+/+}$ and $Reln^{-/-}$ mice. The leptomeninges were removed to reduce fibroblast contamination. Methods to prepare OEC primary cultures were adapted from *Ramón-Cueto et al., 2000* and follow those reported by *Runyan and Phelps, 2009* for mouse OECs and *Khankan et al., 2015* for rat OECs. For scRNA-seq, this protocol was replicated on two separate dates and used a total of eight adult female Sprague-Dawley rats. Primary cultures were generated from both olfactory bulbs from each rodent. The first two layers of the olfactory bulb were dissected, isolated, and then washed in Hank's balanced

salt solution (HBSS, Gibco, Rockville, MD) prior to tissue centrifugation at 365 × $g$ for 5 min. The tissue pellet was resuspended in 0.1% trypsin and HBSS without $Ca^{2+}/Mg^{2+}$ (Gibco, Rockville, MD), then placed in a 37°C water bath, and mixed intermittently for 10 min. A mixture of 1:1 Dulbecco's modified eagle medium (DMEM) and Ham's F12 (D/F medium, Gibco) supplemented with 10–15% fetal bovine serum (FBS, Hyclone, Logan, UT) and 1% penicillin/streptomycin (P/S, Gibco; D/F-FBS-P/S medium) was used to inactivate trypsin prior to centrifugation. Dissociated cells were rinsed, centrifuged three times, and plated into 12.5 cm² culture flasks pre-coated with 0.05 mg/ml poly-L-lysine (PLL, Sigma, St. Louis, MO). Cells were maintained at 37°C for 5–8 days and D/F-FBS-P/S medium was changed every 2 days.

Immunopurification was carried out using hydrophobic Petri dishes coated overnight with Biotin-SP-conjugated AffiniPure goat anti-rabbit IgG (1:1000; Jackson ImmunoResearch, West Grove, PA) in 50 mM Tris buffer at 4°C followed by another overnight incubation at 4°C with either mouse anti-Ngfr$^{p75}$ for rat OECs (Key Resources Table, *Chandler et al., 1984*) or rabbit anti-Ngfr$^{p75}$ for mouse OECs (Key Resources Table) in 25 mM phosphate-buffered saline (PBS). Dishes were rinsed three times with 25 mM PBS and treated with a mixture of PBS and 0.5% bovine serum albumin (BSA) for 1 hr at room temperature. Prior to the addition of cells, antibody-treated dishes were washed with PBS and DMEM.

In preparation for immunopanning, two pairs of adult rat olfactory bulbs (for scRNA-seq) or four pairs of mouse bulbs (for western blots) were dissociated with 0.25% trypsin-EDTA at 37°C for 3 min and D/F-FBS-P/S medium was added to inactivate trypsin. Following a medium rinse and centrifugation, resuspended cells were seeded onto pre-treated mouse or rabbit anti-Ngfr$^{p75}$ Petri dishes and incubated at 37°C for 10 min. Unbound rat cells were removed with medium and saved as 'leftover' control cells for scRNA-seq. A cell scraper was used to recover bound cells that were then subjected to a second round of immunopanning in which purified cells from two rats on each experimental day were combined yielding two samples of purified Ngfr$^{p75}$-positive rat OECs and two samples of 'leftover' control cells. All cells for sequencing were replated into PLL-coated flasks. Immunopurified Ngfr$^{p75}$-positive mouse OECs were plated on PLL-coated 4-chamber polystrene-vessel culture slides (BD Falcon, San Jose, CA) or used for western blots. Purified OECs and 'leftover' controls were incubated at 37°C with 5% $CO_2$ for 7–8 days in D/F-FBS-P/S medium supplemented with a mitogen mixture of pituitary extract (20 µg/ml, Gibco) and forskolin (2 µM, Sigma). Media was changed every 2 days and mitogens were withdrawn 2 days prior to harvesting these cells for scRNA-seq.

## Brefeldin-A treatment

Rat OECs that contained GFP could be visualized directly. Primary and purified GFP-expressing OEC cultures were rinsed three times with D/F medium prior to treatment with Brefeldin-A to block protein transport from the endoplasmic reticulum to the Golgi apparatus and thus block secretion. Brefeldin-A (5 µg/ml; Epicenter, Madison, WI) was mixed with D/F medium and added to OEC cultures for 2 hr at 37°C while the control cultures received D/F medium only. Cultures were fixed with cold 4% paraformaldehyde in 0.1 M PB for 15 min at RT, rinsed three times with PBS, and stored in PBS with sodium azide at 4°C until immunostaining.

## Detection of Reelin expression in rat and mouse OEC cultures, tissue sections, and western blots

Cultured rat OECs treated with or without Brefeldin-A were labeled with mouse anti-Reelin G10 and rabbit anti-Ngfr$^{p75}$ (Key Resources Table). Culture slides were rinsed with PBS (0.1 M phosphate buffer; 0.9% NaCl), blocked with 5% normal goat serum for 1 hr and incubated with the appropriate primary antibodies overnight. The following day, slides were rinsed three times with PBS, incubated with species-appropriate Alexa Fluor 594 and/or 647 (1:500, 1:100; Jackson ImmunoResearch) for 1 hr at RT, and then cover slipped with Fluorogel (Electron Microscopy Sciences, Hatfield, PA).

Sagittal sections of the mouse E16.5 olfactory system were cut 40 µm thick, slide mounted, and stored in PB with 0.06% sodium azide. For Reelin and Blbp detection, sections were incubated overnight at RT in goat-anti-Reelin (Key Resources Table), rinsed thoroughly, and incubated with donkey anti-goat 488 (1:100; Jackson Immunoresearch, #705-545-0030). Sections were then incubated in rabbit polyclonal anti-Blbp (Key Resources Table) overnight followed by 1 hr in donkey anti-rabbit 555 (1:800; Life Technologies, #A31572) and coverslipped with Fluorogel. Confocal images were obtained

from a Zeiss Laser Scanning Microscope (LSM800) with solid-state lasers 488 and 561 nm for double-labeled images.

## Tissue harvesting, protein isolation, and western blotting

Olfactory bulbs from adult $Reln^{+/+}$ and $Reln^{-/-}$ mice and ONLs from adult Sprague-Dawley rats were freshly dissected as described for primary OEC cultures. Olfactory tissue was collected, placed on ice, and homogenized in Ripa lysis buffer plus protease inhibitor cocktail (Sigma). Protein concentrations of brain and olfactory extracts were determined using the RC DC Protein Assay kit (Bio-Rad, Hercules, CA) as described (*Miller et al., 2008*). WCL and CM were obtained from purified OEC cultures maintained in D/F medium. CM was collected from OEC cultures before cells were incubated with Ripa lysis buffer and collected with a cell scraper. Protein samples were stored at –80°C.

Protein samples (50 μg) were heated to 90–100°C for 3 min, resolved on 10% SDS–PAGE or 4–15% mini-protean TGX gels (Bio-Rad) run in Tris-Glycine-SDS (TGS) buffer (Bio-Rad) at 60 V, and then transferred onto 0.22 μm PVDF/nitrocellulose membranes (Bio-Rad) in TGS with 20% methanol at 25 V overnight at 4°C. Precision plus protein standards in dual color (Bio-Rad) were used to determine molecular weight. Membranes were cut, blocked with 5% non-fat dry skimmed milk (Bio-Rad) in Tris-buffered saline containing 0.1% Tween-20 (TBST) for 1 hr at room temperature on a shaker, and then incubated overnight at 4°C TBST plus 2.5% milk containing anti-Reelin G10 or loading control mouse anti-GAPDH antibodies (Key Resources Table). Following primary antibody incubation, blots were washed three times in TBST for 10 min while shaking and then probed with horseradish peroxidase-conjugated secondary anti-mouse IgG (1:2000 for Reelin, 1:5000 for GAPDH) in TBST. Immunoblots were developed using a chemiluminescence HRP detection kit (GE Healthcare, Piscataway, NJ,) and imaged with a Typhoon scanner (GE Healthcare).

For quantification, ImageJ software (NIH) was used to analyze the densitometric data. Western blot images at 400, 300, and 150 kDa resolution were converted to grayscale followed by manually defining a Region of Interest (ROI) frame that captured the entire band in each lane using the 'Rectangular' tool. The area of each selected band was measured by employing the same ROI frame around the band to record the integrated density, 'Grey Mean Value'. Background measurements were similarly quantified, and background subtraction was performed by deducting the inverted background from the inverted band value. For relative quantification, target protein bands were normalized to the corresponding loading control (GAPDH) to derive normalized protein expression (fold change). Band intensities were quantified in triplicate for each sample. Data were analyzed with the Mann–Whitney $U$ test to compare normalized protein expression between the $Reln^{-/-}$ group and the other groups. A one-sided p-value was calculated to test the hypothesis that protein expression levels in the other groups are greater than those in the $Reln^{-/-}$ group. Statistical significance was determined at p < 0.05. Analysis was performed using GraphPad Prism (version 9).

## Single-cell RNA-sequencing

### Cell collection and culture

Flasks with purified OECs and leftover cells were rinsed with HBSS and dissociated with trypsin-EDTA as above. Cells were washed in D/F-FBS-P/S medium and further resuspended with siliconized pipettes. Media and cells were washed and centrifuged at 365 × $g$ for 10 min, and resuspended. After the final wash, cells for sequencing were resuspended in PBS + BSA in volumes of 1200 cells/μl. On each of two different culture days, four flasks of cells (two purified OECs and two 'leftovers') were generated for scRNA-seq, yielding eight samples in total (four purified OECs and four 'leftovers'). To confirm our scRNA-seq results directly with immunocytochemistry, we plated extra cells from the second preparation on PLL-coated coverslips in 24-well plates with 500 μl D/F-FBS-P/S medium. Medium was changed at 2 days, cultures were fixed at 3 days in cold 4% paraformaldehyde for 20 min, washed 3× with PB and stored in PB with 0.06% sodium azide.

scRNA-seq was performed using the 10X Genomics Chromium scRNA-seq system which incorporates microfluidic capture-based encapsulation, barcoding, and library preparation (10X Genomics, Pleasanton, CA). About 10,000 cells per OEC or leftover preparation were loaded into a Chromium Chip B along with partitioning oil, reverse transcription reagents, and a mix of hydrogel beads containing 3,500,000 unique 10X barcodes. Paired-end sequencing was performed on a Novaseq S4 system, using the v3 Illumina platform, at 20–50 K reads/cell. The paired-end reads were processed

using the Cell Ranger pipeline and *Rattus norvegicus* (Rnor_6.0) from Ensembl as the reference genome (*Yates et al., 2020*) to produce a gene expression matrix. Among the eight samples, one OEC sample showed a low fraction of reads and RNA content in cells, which may indicate high level of ambient RNA (*Figure 2—figure supplement 2*). This sample was removed from further analysis, leaving three OEC samples and four leftover samples for downstream analysis.

## scRNA-seq data quality control, normalization, and integration

Gene expression matrix from each of the seven remaining samples that passed quality control was loaded into the Seurat R package ver 3.0 (*Stuart et al., 2019*). Cells were called based on the number of genes (200–5000), unique molecular identifiers (5000–20,000), and percent of mitochondrial genes (<20%; *Figure 2—figure supplement 2*). After quality control, log normalization was performed within each sample using *NormalizeData* function with default parameters. Top 2000 variable genes were selected using *FindVariableFeatures*. The seven samples were integrated together with *FindIntegrationAnchors* and *IntegrateData* functions which incorporate canonical correlation analysis to align cells with similar transcriptomic patterns across samples.

## Cell clustering and cell type identification

The integrated Seurat object was used for the principal component analysis. The top 15 PCs were used to construct the k-nearest neighbor graph, followed by Louvain algorithm to cluster cells based on similar gene expression patterns. Cell clusters were visualized using tSNE plots.

To assign cell type identities to each cluster, marker genes of each cluster were found using *FindAllMarkers* with average log fold change >0.5 and minimum percent difference >0.25. Cell cluster marker genes were subsequently compared to known OEC cell type markers curated from previous studies that used different species, sex, age, dissection areas, and cell preparation methods (*Durante et al., 2020*; *Tepe et al., 2018*). Additional marker genes for fibroblasts and multiple glial cell types including astrocytes, oligodendrocytes, and microglia were also used to compare with those of the cell clusters. Cell clusters whose marker genes match known cell type markers were labeled with the corresponding cell type identity. Gene expression heatmaps of top markers were generated to visualize the distinct expression patterns of the marker genes for different cell clusters.

## OEC subtype identification

A subset of cells expressing well-known OEC markers (e.g., $Ngfr^{p75}$, $S100\beta$, and $Sox10$) was labeled as OECs and then selected for further subclustering analysis to identify potential OEC subtypes. To do this, OECs from the original Seurat object were extracted using the *subset* function and the top 10 PCs of each OEC subset were used for clustering analysis and tSNE visualization. Marker genes for each OEC subcluster were identified using the *FindAllMarkerGene* function. Pseudotime trajectory analysis across the OEC subtypes was performed using Slingshot version 1.8.0 (*Street et al., 2018*).

## Pathway enrichment analysis

To annotate the biological functions of the marker genes of individual cell clusters or OEC subclusters, we carried out pathway enrichment analysis using EnrichR ver 3.0, where KEGG, Reactome, and GO Biological Process were used as reference pathway databases (*Chen et al., 2013a*). Top pathways were ranked by multiple-testing adjusted p-values.

## Potential cell–cell ligand–receptor interactions across OEC subtypes

In order to characterize potential interactions mediated by secreted ligand–receptor pairs between subtypes of OECs, we implemented NicheNet (*Browaeys et al., 2020*), which curates ligand–receptor interactions from various publicly available resources. The NicheNet ligand–receptor model was integrated with markers from each OEC subtype to find potentially activated ligand–receptor pairs. Differentially expressed ligands identified by markers from source OEC subtypes and activated receptors identified by NicheNet in target OEC subtypes were used to assess ligand-receptor interaction. Ligand–receptor interaction networks between cell types were visualized using Cytoscape (*Shannon et al., 2003*).

## Confirmation of OEC gene expression in vitro and in vivo

Immunocytochemistry experiments to confirm gene expression revealed by scRNA-seq were conducted on extra cultured OECs and 'leftover' controls which were replated and cultured for 3 additional days. OECs samples were treated with the following primary antibodies against OEC marker genes identified from scRNA-seq with experimental parameters recorded in the Key Resources Table: (1) polyclonal and monoclonal anti-Blbp, (2) anti-N-Cadherins, (3) anti-green fluorescent protein, (4) anti-Ki67, (5) anti-L1, (6) anti-NCAM, (7) monoclonal anti-Ngfr[p75], (8) anti-S100β, and (9) anti-SOX10. Prior to adding the primary antibodies, the cells were incubated in 0.3% Triton X-100 detergent in PBS. Cells were blocked in 5% normal donkey serum (Jackson ImmunoResearch Laboratories, # 017-000-121) with 0.3% Triton X before adding the primary antibodies and left to incubate at RT overnight on the rotator. Following multiple washes, species-appropriate secondary antibodies with fluorescence for 488, 555, and 647 were used to visualize the immunoreaction (1:200 to 1:500, Jackson ImmunoResearch Labs donkey-anti-chicken-488, # 703-545-155; donkey-anti-rabbit-647, # 711-605-152; donkey-anti-mouse-647, # 715-605-150; Life Technologies, donkey-anti-mouse-555, # A31570; donkey-anti-rabbit-555, # A31572). Cultures were counterstained with Hoechst (Bis-benzimide, 1:500, Sigma-Aldrich, # B2261) and coverslipped with EverBrite medium (Biotium, # 23001).

To examine Ctgf expression in the spinal cord lesion site, we processed 1 slide per animal with ~6 equally spaced sagittal sections (25 µm thick) throughout spinal cords from the *Khankan et al., 2016* study. Our aim was to assess if transplanted OECs (*n* = 4 rats) and transplanted fibroblasts (*n* = 3 rats) express Ctgf in the injury site.

Experiments were carried out to confirm gene expression of OEC subtype markers in the 8- to 10-week-old olfactory system of rats. Soft tissues and the jaw were removed from fixed heads that were decalcified in undiluted formic acid bone decalcifier (Immunocal, Decal Chemical Corps, Tallman, NY) for 12 hr, washed, and then infiltrated with increasing concentrations of 5–20% sucrose for 24 hr. Heads in 20% sucrose were heated in a 37°C oven and then placed in a gelatin sucrose mixture for 4 hr, embedded in warm gelatin sucrose in a plastic mold, allowed to gel, frozen on dry ice and stored at –70°C. Heads were sectioned coronally at 25 µm thick, and mounted in series on 16 glass slides. Several antibodies from each of the four OEC clusters were chosen to determine their distribution in olfactory bulb sections. Primary antibodies used are listed in the Key Resources Table, together with the experimental parameters used. For most antibodies, the olfactory bulb sections first underwent a heat-induced antigen retrieval step with 10 mM citric acid at pH 6 for 5 min and then were transferred to 1.5% NDS and 0.1% Triton X-100 in TBS-BSA (0.1 M Tris; 1.4% NaCl; 0.1% BSA) for 1 hr followed by overnight incubation in primary antibody at RT. Following multiple washes, species-appropriate secondary antibodies with fluorescence for 488 and 555 were used to visualize the immunoreactivity.

For experiments carried out using a Tyramide Signal Amplification (TSA) kit, sections were first incubated in 1% hydrogen peroxide and 0.1% sodium azide in PBS for 30 min followed by 10% NDS and 0.1% Triton X-100 in PBS for 1 hr. Next, the sections were incubated with Avidin-Biotin solution (Vector Laboratories; kit #SP-2001) before an overnight incubation with the primary antibody at RT. The next day, sections were washed in PBS followed by a 1-hr incubation with biotinylated horse anti-goat IgG (1:1000; Vector Laboratories; #BA-9500), biotinylated donkey anti-rabbit IgG (1:1000; Jackson Immunoresearch; #711-065-152), or biotinylated donkey anti-mouse IgG (1:1000; Jackson Immunoresearch; #715-065-150) in TSA-specific blocking buffer (TNB; 0.1 M Tris-HCl; 0.15 M NaCl; 0.5% Blocking reagent; PerkinElmer; #FP1020). Sections were then washed with TNT buffer (0.1 M Tris-HCl; 0.15 M NaCl; 0.05% Tween) followed by a 1-hr incubation with streptavidin-conjugated horseradish peroxidase (1:1000; PerkinElmer; #NEL750001EA) in TNB, and a 5-min incubation with TSA Plus Fluorescein (1:150; PerkinElmer; #NEL741001KT) or TSA Plus Cyanine 3 (Cy3; 1:100; Perkin-Elmer; #NEL744001KT). Sections were imaged on an Olympus AX70 microscope with a Zeiss AxioCam HRcRv.2 camera. Images were transferred to Adobe Photoshop for assembly, cropping, and adjustment of brightness/contrast to create the compiled figures.

## Morphological analyses of Ki67 OEC subtypes

To determine if OEC progenitor cells marked with Ki67 immunoreactivity have a distinctive morphology, purified and fixed OEC cultures from four rats were processed with anti-Ngfr[p75], anti-Ki67 and counterstained with Hoechst (Bis-benzimide, 1:500, Sigma-Aldrich, #B2261). Images were acquired from 7 to 10 randomly selected fields/sample using an Olympus AX70 microscope and Zen image processing

and analysis software (Carl Zeiss). We distinguished the larger, flat 'astrocyte-like' OECs from the smaller, fusiform 'Schwann cell-like' OECs, and recorded their expression of Ngfr$^{p75}$ and Ki67. Cell counts from each field were averaged per rat and then averaged into a group mean ± SEM. A Student $t$-test was conducted to compare the effect of Ngfr$^{p75}$-labeled cell morphology and the proliferative marker Ki67. Statistical significance was determined by $p < 0.05$.

## Acknowledgements

This work was supported by the National Institute of Neurological Disorders and Stroke: RO1 NS076976 (PEP), R01 NS111378 (XY), R01 NS117148 (XY), and by NICHD of the National Institutes of Health under award number P50HD103557 (confocal microscope resources). Additional support came from the UCLA QCBio Collaboratory Postdoc Fellowship (SMH), Eureka fellowship (RRK), MARC U*STAR fellowship T34 GM008563 (GJ), and UCLA Graduate Year Fellowship (KLID). We thank Drs. Julie Miller and Stephanie White for their generosity and considerable assistance with western blots, Drs. Hui Zhong and VR Edgerton for their long-term collaboration on spinal cord injury studies, and Sun Young Lee for excellent technical assistance.

## Additional information

### Funding

| Funder | Grant reference number | Author |
|---|---|---|
| National Institute of Neurological Disorders and Stroke | RO1 NS076976 | Patricia E Phelps |
| National Institute of Neurological Disorders and Stroke | R01 NS111378 | Xia Yang |
| National Institute of Neurological Disorders and Stroke | R01 NS117148 | Xia Yang |
| UCLA QCBio Collaboratory Postdoc Fellowship | | Sung Min Ha |
| Eureka Fellowship | | Rana R Khankan |
| MARC U*STAR Fellowship | T34 GM008563 | Giovanni Juarez |

The funders had no role in study design, data collection, and interpretation, or the decision to submit the work for publication.

### Author contributions

Patricia E Phelps, Conceptualization, Data curation, Supervision, Funding acquisition, Validation, Investigation, Writing – original draft, Project administration, Writing – review and editing; Sung Min Ha, Data curation, Software, Formal analysis, Writing – review and editing; Rana R Khankan, Conceptualization, Data curation, Formal analysis, Investigation, Writing – original draft, Writing – review and editing; Mahlet A Mekonnen, Giovanni Juarez, Validation, Methodology, Writing – review and editing; Kaitlin L Ingraham Dixie, Data curation, Validation, Methodology; Yen-Wei Chen, Software, Formal analysis; Xia Yang, Conceptualization, Data curation, Software, Formal analysis, Supervision, Funding acquisition, Methodology, Writing – original draft, Writing – review and editing

### Author ORCIDs

Patricia E Phelps https://orcid.org/0000-0003-0735-5341
Sung Min Ha https://orcid.org/0000-0003-3945-8329
Xia Yang https://orcid.org/0000-0002-3971-038X

### Ethics

All of the experimental procedures used on animals were approved by the Chancellor's Animal Research Committee at UCLA (996-0135-15) and carried out in accordance with the National

Institutes of Health guidelines. An overdose of ketamine–xylazine or pentobarbital was used for euthanasia before either the rat olfactory bulbs were removed or the rats were perfused with fixative.

Reviewer #1 (Public review): https://doi.org/10.7554/eLife.95629.4.sa1
Reviewer #2 (Public review): https://doi.org/10.7554/eLife.95629.4.sa2
Author response https://doi.org/10.7554/eLife.95629.4.sa3

## Additional files

### Supplementary files
Supplementary file 1. Differentially enriched marker genes for fibroblasts, microglia, and olfactory ensheathing cells (OECs) identified by scRNA-seq. The list of genes that are significantly enriched in each major cell type (fibroblast, microglia, or OEC) based on single-cell RNA-sequencing of immunopurified OECs and 'leftover' control cultures. For each gene, the table shows the nominal p-value, the multiple-testing adjusted p-value, the log2 fold change (relative to all other cell types), and the percentage of cells expressing that gene within the indicated cell type versus all other cell types. Genes are grouped by the cell type in which they are most significantly enriched.

Supplementary file 2. Differentially enriched marker genes for each olfactory ensheathing cell (OEC) subcluster identified by scRNA-seq. The list of genes that are significantly enriched in each of the five OEC subclusters identified by single-cell RNA-sequencing. The columns show the nominal p-value, the multiple-testing-adjusted P value, the log2 fold change (relative to all other OEC subclusters), and the OEC subcluster (0–4) in which each gene was most strongly enriched. Genes are grouped based on the subcluster where their expression is highly specific, providing a unique transcriptional signature for each OEC subtype.

MDAR checklist

### Data availability
Data availability: Raw scRNA-seq data and expression matrix are available at GEO (accession number GSE215247). The scRNA-seq results also can be viewed interactively at the Single Cell Portal. OEC versus Leftovers: https://singlecell.broadinstitute.org/single_cell/study/SCP1237/leftovers-and-oecs. OEC subclusters: https://singlecell.broadinstitute.org/single_cell/study/SCP1489/oec-subcluster-batch-corrected.

The following dataset was generated:

| Author(s) | Year | Dataset title | Dataset URL | Database and Identifier |
|---|---|---|---|---|
| Phelps PE, Ha SM, Yang X | 2022 | Single cell RNA-sequencing reveals five subtypes of olfactory bulb-derived olfactory ensheathing cells that secrete Reelin | https://www.ncbi.nlm.nih.gov/geo/query/acc.cgi?acc=GSE215247 | NCBI Gene Expression Omnibus, GSE215247 |

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

# Appendix 1

## Top 20 marker genes based on log2 fold change in each of the 5 OEC clusters revealed by scRNA-seq

The top 20 most highly enriched genes for each of the five OEC subclusters identified by single-cell RNA-sequencing. For each gene, its fold-change value (log2) and a concise summary of its known or putative functions are provided, along with references to its expression patterns in glial and other cell types. This format highlights the most distinctive and potentially significant molecular markers that define the diversity of OEC subtypes.

### Cluster 0 – Genes listed in descending order

| OEC sub-cluster genes (Fold change) | Name and Function | Where is it expressed? | References |
|---|---|---|---|
| 1 Pmp22 (1.427) | Peripheral myelin protein 22 is a myelin associated gene | Schwann cell myelin | *Notterpek et al., 1999*; *Roet et al., 2011*; *Salzer, 2015* |
| 2 Ccdc80 (1.166) | Coiled-coil Domain containing 80, secreted, part of Hedgehog pathway, may regulate vascular function | Zebrafish somites, axonal pathfinding | *Brusegan et al., 2012*; *Della Noce et al., 2015*; *Sasagawa et al., 2016* |
| 3 Gldn (1.130) | Gliomedin is a required ECM protein that initiates assembly of PNS nodes of Ranvier | Schwann cell surface | *Colognato and Tzvetanova, 2011*; *Eshed et al., 2005* |
| 4 Ccn3/Nov (1.165) | Nephroblastoma over-expressed, matricellular protein with proregenerative function, Ccn3 upregulates chemokines CCL2 and CXCL1 | Secreted by astrocytes a&nd regulatory T cells; promotes oligodendrocyte differentiation and myelination | *de la Vega Gallardo et al., 2020*; *Le Dréau et al., 2010*; *Malik et al., 2015* |
| 5 Col11A1 (1.049) | Collagen, type XI, alpha 1 ECM protein | Schwann cells | *Zhang et al., 2021* |
| 6 Serpinf1 (1.035) | Serine protease inhibitor, Neurotrophic functions, enhances axon outgrowth | Dorsal root ganglion axons in vitro | *Roet et al., 2011*; *Roet et al., 2013* |
| 7 Fbln2 (1.026) | Fibulin2; mediates the TGF-β1 signaling in adult neural stem cells | Astrocytes in adult neurogenesis (SVZ) | *Radice et al., 2015* |
| 8 Nefm (0.976) | Neurofilament medium | Glia, neurons | *Roet et al., 2011* |
| 9 Ccn2 (0.91) | Connective tissue growth factor, secreted CTGF regulates growth factors, cytokines and MMPs | Astrocytes, Schwann cells | *Jun and Lau, 2011*; *Lamond and Barnett, 2013*; *Malik et al., 2015*; *Roet et al., 2011* |
| 10 RGD1305645 (0.87) | Ecrg4 augurin precursor, esophageal cancer-related gene 4 protein | Uncharacterized in nervous system | Rat Genome Database https://rgd.mcw.edu |
| 11 Nqo1 (0.849) | NAD(P)H dehydrogenase, quinone 1; protect CNS cells against oxidative stress | Bergmann glia, astrocytes,oligodendrocytes | *Roet et al., 2011*; *Stringer et al., 2004* |
| 12 Marcks (0.839) | Myristoylated alanine-rich C-kinase substrate protein; actin-cross-linking protein regulates radial glia function | Astrocytes, oligodendrocytes | *Artegiani et al., 2017*; *Roet et al., 2011*; *Weimer et al., 2009* |
| 13 Prss23 (0.814) | Novel vascular protease required for endothelial-to-mesenchymal transition | Cardiovascular valve development | *Chen et al., 2013b*; *Roet et al., 2011* |
| 14 Rarres2 (0.791) | Chemerin or Retinoic acid receptor responder 2, an inflammatory chemokine; under neuroendocrine control | Ventricular ependymal cells, tanycytes | *Helfer et al., 2016* |
| 15 Cck (0.775) | Cholecystokinin, gut hormone and neuropeptide | Peripheral nerves, neurons | *Rehfeld, 2017* |

*Continued on next page*

*Continued*

| OEC sub-cluster genes (Fold change) | Name and Function | Where is it expressed? | References |
|---|---|---|---|
| 16 Fst (0.767) | Follistatin; plays a role in myelination and controls activin | Olfactory bulb, ONL; Areas of adult neurogenesis, | *Ogawa et al., 2020* |
| 17 Tagln (0.767) | Transgelin, a TGFβ-inducible gene that functions as an actin-crosslinking protein | Vascular endothelium, smooth muscle, neural crest stem cells | *Elsafadi et al., 2016; Roet et al., 2011* |
| 18 Fn1 (0.728) | Fibronectin-1; major component of ECM | Perivascular fibroblasts, macrophages, reactive astrocytes | *Colognato and Tzvetanova, 2011; Zhu et al., 2015* |
| 19 Cd200 (0.728) | CD200 binds to and activates its receptor (CD200R) on microglial cells, have immunoregulatory functions | Neurons, astrocytes, oligodendrocytes | *Manich et al., 2019* |
| 20 Dpysl3 (0.691) | Dihydropyrimidinase-like 3; important for microglia activation, migration and phagocytic ability | Ameboid microglia | *Manivannan et al., 2013* |

## Cluster 1– Genes listed in descending order

| OEC sub-cluster genes (Fold change) | Name and Function | Where is it expressed? | References |
|---|---|---|---|
| 1 Pcsk1 (1.616) | Preprohormone convertase 3 or Neuroendocrine convertase 1 | Schwann cells | *Marcinkiewicz et al., 1998; Roet et al., 2011* |
| 2 Gal (1.608) | Galanin, small neuropeptide member of corticotropin-releasing factor family; regulates oligodendrocyte survival | Neurons, neural progenitor cells in areas of adult neurogenesis, oligodendrocytes | *Cordero-Llana et al., 2014; Gresle et al., 2015; Roet et al., 2011; Ubink et al., 2003* |
| 3 Atf3 (1.234) | Activating transcription factor 3, a pro-regenerative intrinsic factor enhances PNS axon regeneration | Schwann cells, injured dorsal root ganglia, axotomized axons | *Hunt et al., 2004; Mahar and Cavalli, 2018; Shortland et al., 2006; Zhang et al., 2021* |
| 4 Ucn2 (1.223) | Urocortin-2 protein member, neuropeptide involved in stress response and axonal outgrowth, corticotropin-releasing factor family | Hypothalamic neurons, motor neurons in brain stem, perisynaptic Schwann cells | *D'Este et al., 2022; Reyes et al., 2001; Roet et al., 2011* |
| 5 Crabp2 (1.142) | Cellular retinoic binding protein 2 transports retinoic acid to the nuclear receptors | Neural crest-derived cranial ganglia, olfactory placode, olfactory epithelium | *Cai et al., 2012* |
| 6 Gap43 (1.137) | Growth associated protein 43, transiently expressed in growing axons and after axotomy | Growing axons, Schwann cell precursors | *Jacobson et al., 1986; Sensenbrenner et al., 1997; Takeoka et al., 2011* |
| 7 Btc (1.079) | Betacellulin, part of Epidermal Growth Factor family, a ligand for EGFR and a growth factor | Myelinating Schwann cells | *Roet et al., 2011; Vallières et al., 2017* |
| 8 LOC108348064 (1.078) | Aph-1 homolog B, gamma secretase subunit, participates in Notch pathway | Predicted location at synapse and transport vesicle | Rat Genome Database https://rgd.mcw.edu |
| 9 Chrdl2 (1.066) | Chordin-like 2, a novel secreted BMP binding inhibitor | Chondrocytes, regulates dorso-ventral patterning | *Branam et al., 2010; Nakayama et al., 2004* |
| 10 Cast (1.038) | Calpastatin, endogenous calpain inhibitor, key regulator of axon survival post injury and during development | Broadly expressed in CNS & PNS, including Purkinje neurons and retinal ganglion cells. | *Yang et al., 2013* |
| 11 Il11 (0.961) | Interleukin-11, secreted by astrocytes, enhances oligodendrocyte survival, maturation and myelin formation | Astrocytes | *Roet et al., 2011; Zhang et al., 2006* |

*Continued on next page*

*Continued*

| OEC sub-cluster genes (Fold change) | Name and Function | Where is it expressed? | References |
|---|---|---|---|
| 12 Ngfr (0.96) | Nerve growth receptor-p75 enhances axonal growth and remyelination | Schwann cells | *Roet et al., 2011*; *Tomita et al., 2007* |
| 13 Nes (0.953) | Nestin, an intermediate filament associated with neural stem cells | Neuroepithelial stem cells, radial glia, adult neural crest-related stem cells | *Cattaneo and McKay, 1990*; *Widera et al., 2011* |
| 14 S100ß (0.918) | S100ß is a calcium binding protein that increases after injury and supports neurogenesis | Schwann cells, astrocytes, NG2 cells, APC +oligodendrocytes in olfactory bulb | *Su et al., 2021*; *Villarreal et al., 2014* |
| 15 Vcan (0.903) | Versican, a chondroitin sulfate proteoglycan | Astrocytes, late oligodendrocyte progenitors | *Anderson et al., 2016*; *Perlman et al., 2020* |
| 16 Bdnf (0.901) | Brain-derived neurotrophic factor, axonal growth and regeneration, dendritic growth | Schwann cells | *Gordon, 2009*; *Guérout et al., 2010*; *Meyer et al., 1992*; *Runyan and Phelps, 2009* |
| 17 Stim1 (0.884) | Stromal interaction molecule, $Ca^{2+}$ sensor in ER that mediates store-operated $Ca^{2+}$ entry (SOCE) | Astrocytes, microglia | *Michaelis et al., 2015*; *Müller et al., 2014* |
| 18 Serpinb6a (0.874) | Serpin family B member 6 a, part of serine protease inhibitor complex, loss causes hearing deficit | Auditory related neurosensory epithelium | Rat Genome Database https://rgd.mcw.edu |
| 19 Hspb1 (0.858) | Heat shock protein b1, small heat shock protein | Highly expressed in glial cells | *López-González et al., 2014* |
| 20 LOC102549-726 (0.811) | Uncharacterized | Uncharacterized | Rat Genome Database https://rgd.mcw.edu |

## Cluster 2 – Genes listed in descending order

| OEC sub-cluster genes (Fold change) | Name and Function | Where is it expressed in progenitor cells? | References |
|---|---|---|---|
| 1 Stmn1 (2.502) | Stathmin 1, microtubule destabilizing protein antagonizes differentiation of oligodendrocytes | Areas of adult neurogenesis-SVZ, Oligodendrocyte progenitors, astrocytes | *Boekhoorn et al., 2014*; *Liu et al., 2003* |
| 2 Cenpf (2.265) | Centromere protein F, associates with centromere-kinetochore complex | In nuclear matrix during G2 phase in interphase, Schwann cells | *Whitfield et al., 2002*; *Zhang et al., 2021* |
| 3 Prc1 (2.118) | Polycomb repressive complex 1, repressive epigenetic mechanism that controls stem cell self-renewal or differentiation | Widely expressed in neurons and glia, proliferative globose basal cells in olfactory epithelium | *Goldstein et al., 2018* |
| 4 Hmgb2 (2.099) | High mobility group box 2, secreted proinflammatory cytokine | Expressed in the nervous system during embryogenesis | *Lee et al., 2014* |
| 5 Pttg1 (1.971) | Pituitary tumor transforming gene 1 | Regulates cell cycle M/G1, sister chromatid cohesion | *Whitfield et al., 2002* |
| 6 Ube2c (1.966) | Ubiquitin conjugating enzyme E2 C | Regulates cell cycle G2/M phase | *Clark et al., 2019*; *Whitfield et al., 2002* |
| 7 Cdk1 (1.951) | Cyclin dependent kinase 1 binds to Cyclin B1 to enter mitosis | Regulation of cell cycle, mitosis | *Martinsson-Ahlzén et al., 2008*; *Morgan, 1997* |
| 8 Cenpw (1.895) | Component of CENPA-Nucleosome-associated complex required in progenitor proliferation | Required during mitosis for kinetochore formation and centriole splitting | *Aygün et al., 2021* |

*Continued on next page*

*Continued*

| OEC sub-cluster genes (Fold change) | Name and Function | Where is it expressed in progenitor cells? | References |
|---|---|---|---|
| 9 Cdkn3 (1.876) | Cyclin dependent kinase inhibitor 3 interacts with Cdk2 and prevents its activation | Cell cycle regulation, G1/S transition | *Nalepa et al., 2013* |
| 10 Top2a (1.855) | DNA topoisomerase II alpha | Cell cycle regulation, peaks in G2 | *Whitfield et al., 2002* |
| 11 Tpx2 (1.855) | TPX2 microtubule nucleation factor, part of acentrosomal MT organizing center | Localized to centrosome and neurite shaft | *Chen et al., 2017* |
| 12 Cks2 (1.828) | CDC28 protein kinase regulatory subunit 2, counteracts Cks1 and stabilizes p27 | Active from late G1 to mitosis, regulator of stem cell maintenance and differentiation | *Frontini et al., 2012*; *Martinsson-Ahlzén et al., 2008* |
| 13 LOC100359539 (1.808) | Ribonucleotide reductase M2 polypeptide | Involved in deoxyribonucleotide biosynthetic process | Rat Genome Database https://rgd.mcw.edu |
| 14 Spc24 (1.706) | SPC24 component of NDC80 kinetochore complex | Kinetochore-microtubule attachments and spindle checkpoint activity | *McCleland et al., 2004* |
| 15 Dut (1.614) | Deoxyuridine triphosphatase (dUTPase), prevents uracil misincorporation into DNA | Essential enzyme for nucleotide metabolism and DNA repair | *Ye et al., 2019* |
| 16 Mki67 (1.613) | Ki-67 is an early progenitor marker of proliferation, component of mitotic cell periphery | Nuclei of proliferating cells, Schwann cell subtype 4 | *Artegiani et al., 2017*; *Zhang et al., 2021* |
| 17 LOC100360316Hmgn2-ps4 (1.585) | High mobility group nucleosomal binding domain 2 | Chromatin in the nucleus | Rat Genome Database https://rgd.mcw.edu/ |
| 18 Ckap2 (1.573) | Cytoskeleton associated protein 2, stabilizes microtubules | Maintains microtubule nucleation sites, regulates cell division | *Case et al., 2013* |
| 19 Fam111a (1.512) | Family with sequence similarity 111 member A, cell-cycle regulated | Mediates effects of protein obstacles on DNA replication forks | *Kojima et al., 2020* |
| 20 Cdca8 (1.493) | Cell division cycle associated 8, part of a chromosomal passenger complex | Required for chromatin-induced microtubule stabilization and spindle formation | *Sampath et al., 2004* |

## Cluster 3 – Genes listed in descending order

| OEC sub-cluster genes (Fold change) | Name and Function | Where is it expressed? | References |
|---|---|---|---|
| 1 C1qa (3.981) | Complement C1q subcomponent a, part of the innate immune system | Macrophages, microglia | *Durante et al., 2020*; *Zhang et al., 2014* |
| 2 Tyrobp (3.944) | TYRO protein tyrosine kinase binding protein (DAP 12), a microglial membrane adaptor that mediates microglial upregulation and neuropathic pain | Microglia | *Guan et al., 2016*; *Zhang et al., 2014* |
| 3 C1qb (3.905) | Complement C1q subcomponent b | Macrophages, microglia | *Durante et al., 2020*; *Zhang et al., 2014* |
| 4 Apoe (3.877) | Apolipoprotein E is a cholesterol transport gene with many functions, binds to low-density lipoprotein receptor gene family | Astrocytes, cultured microglia | *Cadiz et al., 2022*; *Herz and Chen, 2006* |
| 5 Fcer1g (3.665) | Fc receptor, IgE, high-affinity I, gamma polypeptide. Activates microglial phagocytosis pathways | Microglia, astrocytes, oligodendrocytes, neurons | *Sierksma et al., 2020*; *Zhang et al., 2014* |

*Continued*

| OEC sub-cluster genes (Fold change) | Name and Function | Where is it expressed? | References |
|---|---|---|---|
| 6 Cst3 (3.499) | Cystatin C protein | Microglia | *Zhang et al., 2014* |
| 7 Tmem176b (3.491) | Transmembrane protein 176b | Immature monocytes and dendritic cells, Schwann cells | *Condamine et al., 2010*; *Zhang et al., 2021* |
| 8 Anxa3 (3.4) | Annexin A3, calcium-dependent lipid binding protein | Microglia, monocytes | *Smithson and Kawaja, 2010* |
| 9 Tmem176a (3.365) | Transmembrane protein 176a, paralog of Tmem176b | Immature monocytes and dendritic cells | *Condamine et al., 2010* |
| 10 C1qc (3.183) | Complement C1q subcomponent c | Macrophages, microglia | *Cadiz et al., 2022*; *Durante et al., 2020*; *Zhang et al., 2014* |
| 11 Aif1 (3.033) | Allograft inflammatory factor 1, ionized calcium-binding adaptor molecule (Iba1), marker to identify microglia | Macrophages, microglia | *Artegiani et al., 2017* |
| 12 Csf1r (2.559) | Colony stimulating factor 1 receptor, binds to colony stimulating factor 1, is upregulated after injury | Surface of microglia | *Guan et al., 2016*; *Roet et al., 2011*; *Zhang et al., 2014* |
| 13 Ctsz (2.436) | Cathepsin Z, lysosomal cysteine proteinase, enables cysteine-type endopeptidase activity | Microglia | *Zhang et al., 2014*; Rat Genome Database |
| 14 Laptm5 (2.338) | Lysosomal protein transmembrane 5 | Microglia | *Artegiani et al., 2017*; *Roet et al., 2011* |
| 15 Ctsb (2.288) | Cathepsin B, lysosomal cysteine proteinase involved in phagocytosis | Microglia | *Grubman et al., 2021*; *Zhang et al., 2014* |
| 16 Lyz2 (2.262) | Lysozyme C type 2, lysosomal gene | Microglia | *Cadiz et al., 2022*; *Roet et al., 2011* |
| 17 Rgs10 (2.237) | Regulator of G protein signaling 10, in RGS superfamily, selectively interacts with Gαi proteins | Microglia, monocyte-derived dendritic cells | *Almutairi et al., 2020*; *Artegiani et al., 2017* |
| 18 Cd68 (2.134) | Cluster of Differentiation 68, glycoprotein associated with endosomal/lysosomal compartment | Microglia, Macrophages, other mononuclear phagocytes | *Chistiakov et al., 2017* |
| 19 Arhgdib (1.993) | Rho GDP dissociation inhibitor-b | Microglia | *Artegiani et al., 2017* |
| 20 Trem2 (1.908) | Triggering receptor expressed on myeloid cells 2, Trem2 protein interacts with protein produced from Tyrobp gene | Microglia, immature dendritic cells | *Artegiani et al., 2017*; *Zhang et al., 2014* |

## Cluster 4 – Genes listed in descending order

| OEC sub-cluster genes (Fold change) | Name and Function | Where is it expressed? | References |
|---|---|---|---|
| 1 Sidt2 (2.833) | SID1 transmembrane family member 2 binds RNA/DNA, transports nucleic acids to lysosomes for degradation | Brain lysosomes | *Hase et al., 2020*; *Jialin et al., 2010* |
| 2 Ckb (2.687) | Brain-type creatine kinase, fuels ATP-dependent cytoskeletal dynamics, facilitates spreading of cortical actin for motility | Astrocytes, oligodendrocytes, macrophages | *Kuiper et al., 2009* |

*Continued on next page*

*Continued*

| OEC sub-cluster genes (Fold change) | Name and Function | Where is it expressed? | References |
|---|---|---|---|
| **3** Npvf (2.196) | Neuropeptide VF precursor or gonadotropin-inhibitory hormone. | Hypothalamic neurons, gonads, olfactory bulbs-terminal nerve | *Ubuka et al., 2016* |
| **4** Ptn (2.127) | Pleiotrophin, secreted-heparin-binding protein, neurotrophic growth factor for neurite and glial process outgrowth during development and after injury | Microglia, macrophages, astrocytes, pericytes, endothelial cells | *Nikolakopoulou et al., 2019*; *Yeh et al., 1998* |
| **5** C1ql1 (1.928) | Complement C1q like 1, involved in different aspects of synaptic remodeling, synapse elimination on Purkinje cells | Neurons such as cerebellar granule cells, oligodendrocyte precursors | *Eroglu and Barres, 2010*; *Kakegawa et al., 2015*; *Zhang et al., 2014* |
| **6** Igfbp3 (1.924) | Insulin-like growth-factor binding protein-3 reduces apoptosis of pericytes and retinal neurons | Astrocytes, endothelial cells | *Kielczewski et al., 2011*; *Watanabe et al., 2015* |
| **7** Gria2 (1.922) | Glutamate receptor ionotropic AMPA 2 determines channel conductance and $Ca^{2+}$ permeabi-lity, with RNA editing (Q/R site) channel is impermeable to $Ca^{2+}$ | Astrocytes, Oligodendrocyte precursors (OPCs) | *Chen et al., 2018*; *Rusnakova et al., 2013*; *Zhang et al., 2014* |
| **8** Stmn2 (1.842) | Stathmin2 is a tubulin-binding protein that regulates microtubule disassembly | Neural stem cells, neural crest cells, neurons | *He et al., 2016*; *Himi et al., 1994*; *Zhang et al., 2014* |
| **9** Actn1 (1.825) | Actinin, alpha1, cross-links actin filaments to maintain cell scaffolds & mechanical stability | Migrating cells | *Sjöblom et al., 2008* |
| **10** Vgf (1.783) | Secreted nerve growth factor with neuroendocrine functions, promotes oligodendrogenesis | Neurons and neuroendocrine cells | *Alvarez-Saavedra et al., 2016*; *Zhang et al., 2014* |
| **11** Col8a1 (1.646) | Collagen, type 8, alpha1, short chain collagen involved in cell adhesion, migration, & sprouting angiogenesis | Endothelial cells, vasculature, smooth muscle, mast cells | *Illidge et al., 1998* |
| **12** Nacc2 (1.537) | Nacc family member 2, multiple functions such as DNA-binding repressor | Chromatin, nucleolus, mitochondrion | Rat Genome Database https://rgd.mcw.edu |
| **13** Thbs2 (1.472) | Thrombospondin 2, secreted ECM glycoprotein contributes to synaptogenesis during development and repair of blood-brain barrier | astrocytes | *Christopherson et al., 2005*; *Tian et al., 2011* |
| **14** Bgn (1.465) | Biglycan, secreted chondroitin sulfate proteoglycan with neuronal survival-enhancing activity | Meningeal fibroblasts, pericytes | *Junghans et al., 1995*; *Zhang et al., 2014* |
| **15** Mt3 (1.462) | Metallothionein-3, maintains homeostasis of copper and zinc, regulates lysosomal function, associated with growth inhibitory factor | Brain, astrocytes | *Lee et al., 2010* |
| **16** Rarres1 (1.458) | Retinoic acid receptor responder 1. Expression induced by retinoic acid, mainly studied in cancer as a potential tumor suppressor | Subset of stellate and superior cervical sympathetic ganglion neurons (NA1) | *Furlan et al., 2016*; *Mapps et al., 2022*: *Roet et al., 2011* |
| **17** Tubb2b (1.458) | Tubulin b, class 2b, together with a-tubulin form microtubules critical for cell movement or neuronal migration | Neurons, radial glia | *Chakraborti et al., 2016* |
| **18** Tubb3 (1.435) | Tubulin b, class 3, neuronal specific, together with a-tubulin form microtubules critical for migration axonal guidance and transport | Neurons, dorsal root ganglion | *Chakraborti et al., 2016*; *Zhang et al., 2021*; *Zhang et al., 2014* |
| **19** C1qtnf5 (1.413) | C1q tumor necrosis factor 5, secreted membrane-associated protein in RPE, when mutated, destabilizes protein & leads to photoreceptor death | Retinal pigment epithelium (RPE) | *Stanton et al., 2017* |
| **20** Gpm6b (1.345) | Glycoprotein M6b, proteolipid protein family member binds cholesterol and contributes to CNS myelin, in Schwann cell nodes of Ranvier | Oligodendroglia, Schwann cell microvilli, neurons, | *Bang et al., 2018*; *Roet et al., 2011*; *Werner et al., 2013* |

