## [Editor Report · eLife Assessment]

This study unveils **important** data describing cell states of olfactory ensheathing cells, and how these cell states may relate to repair after spinal cord injury. The framework used for characterizing these cells is **solid**. This work will be of interest to stem cell biologists and spinal cord injury researchers.

---

## [Referee Report · Reviewer #1 (Public review)]

The goal of this study was to identify the phenotype of olfactory ensheathing cells (OECs) that have been associated with neural tissue repair, and investigate the properties of these cells that can be used to identify them. OECs modify inhibitory glial scar formation, enabling axon regeneration past the scar border and into the lesion center. Single-cell RNA sequencing revealed diverse subtypes of OECs expressing novel marker genes associated with progenitor, axonal regeneration, repair, and microglia-like functions, suggesting their potential roles in wound healing, injury repair, and axonal regeneration. Additionally, the study identified secreted molecules such as Reelin and Connective tissue growth factor, which are important for neural repair and axonal outgrowth, further supporting the multifunctional nature of OECs in facilitating spinal cord injury recovery. This is an extremely well written and impactful series of experiments from a renowned leader in the field. The experimental questions are timely, with similar therapeutic approaches being prepared for clinical trial. The results address a gap that has persisted in the field for several decades, and one that has asked by many scientists long before technology existed to find answers. This highlights the importance of these experiments and the results reported here. The authors have also included a thoughtful discussion that highlights the importance of their data in the context of prior research. They have carefully interpreted their results and also indicate where additional studies in future work will continue to expand our knowledge of these important cells and their potential use for neural repair.

---

## [Referee Report · Reviewer #2 (Public review)]

Summary

This manuscript explores the transcriptomic identities of olfactory ensheathing cells (OECs), glial cells that support life-long axonal growth in olfactory neurons, as they relate to spinal cord injury repair. The authors show that transplantation of cultured, immunopurified rodent OECs at a spinal cord injury site can promote injury-bridging axonal regrowth. They then characterize these OECs using single-cell RNA sequencing, identifying five subtypes and proposing functional roles that include regeneration, wound healing, and cell-cell communication. They identify one progenitor OEC subpopulation and also report several other functionally relevant findings, notably, that OEC marker genes contain mixtures of other glial cell type markers (such as for Schwann cells and astrocytes), and that these cultured OECs produce and secrete Reelin, a regrowth-promoting protein that has been disputed as a gene product of OECs.

Strengths

This manuscript offers an extensive, cell-level characterization of OECs, supporting their potential therapeutic value for spinal cord injury and suggesting potential underlying repair mechanisms. The authors use various approaches to validate their findings, providing interesting images that show the overlap between sprouting axons and transplanted OECs, and showing that OEC marker genes identified using single-cell RNA sequencing are present in vivo, in both olfactory bulb tissue and spinal cord after OEC transplantation.

Concerns about quantification raised during the review were suitably addressed by the authors.

---

## [Author Response]

The following is the authors’ response to the previous reviews

**Joint Public Reviews:**
SummaryThis manuscript explores the transcriptomic identities of olfactory ensheathing cells (OECs), glial cells that support life-long axonal growth in olfactory neurons, as they relate to spinal cord injury repair. The authors show that transplantation of cultured, immunopurified rodent OECs at a spinal cord injury site can promote injury-bridging axonal regrowth. They then characterize these OECs using single-cell RNA sequencing, identifying five subtypes and proposing functional roles that include regeneration, wound healing, and cell-cell communication. They identify one progenitor OEC subpopulation and also report several other functionally relevant findings, notably, that OEC marker genes contain mixtures of other glial cell type markers (such as for Schwann cells and astrocytes), and that these cultured OECs produce and secrete Reelin, a regrowth-promoting protein that has been disputed as a gene product of OECs.StrengthsThis manuscript offers an extensive, cell-level characterization of OECs, supporting their potential therapeutic value for spinal cord injury and suggesting potential underlying repair mechanisms. The authors use various approaches to validate their findings, providing interesting images that show the overlap between sprouting axons and transplanted OECs, and showing that OEC marker genes identified using single-cell RNA sequencing are present in vivo, in both olfactory bulb tissue and spinal cord after OEC transplantation.ChallengesDespite the breadth of information presented, and although many of the suggestions in the initial review were addressed well, some points related to quantification and discussion of sex differences are not fully addressed in this revision.(1) The request for quantification of OEC bridges is not fully addressed. We note that this revision includes the following statement (page 6): "We note, however, that such bridge formation is rare following a severe spinal cord injury in adult mammals." However, the title of the paper states that olfactory ensheathing cells promote neural repair and the abstract states that "OECs transplanted near the injury site modify the inhibitory glial scar and facilitate axon regeneration past the scar border and into the lesion." Statements such as these make it more crucial to include quantification of OEC bridges, because if single images are shown of remarkable, unusual bridges, but only one sentence acknowledges the low frequency of this occurrence, then this information taken together might present the wrong takeaway to readers.Including some sort of quantification of bridging, whether it be the number of rats exhibiting bridges, the percentage area of OECs near a lesion site, or some other meaningful analysis, would add rigor and clarity to the manuscript.

The short answer to the OEC bridges quantification is that in our last 2 studies combined, we observed bridges in 3/13 OB-OEC-transplanted rats versus 0/16 control rats (p=0.042 by two-sample proportion test; Thornton et al., 2018, Dixie, 2019). In addition to the new data on bridge formation shown in the current manuscript, our previous and most impressive data of serotonergic axons (5-HT-labeled, red) that crossed the entire lesion site is shown below (from Thornton et al., 2018). The image together with Supplemental video 1 (https://ars.els-cdn.com/content/image/1-s2.0-S0014488618302632-mmc1.mp4) show a reconstruction of multiple sections containing serotonergic axons that bridge the injury site in one OEC-transplanted, completely transected rat (1/5 OEC vs. 0/5 fibroblast-transplanted rat). The video also shows retrogradely-labeled Pseudo-rabies virus taken up by a few scattered neurons (green dots) within and above the lesion site, additional evidence suggesting axonal regeneration.

In addition to adding bridge quantification in the Results section, we now discuss quantified results on physiological and anatomical evidence of axon regeneration across the injury site from five of the six large spinal cord injury (SCI) studies conducted by the Phelps and Edgerton laboratories. Our studies used the most difficult SCI model, a complete, thoracic spinal cord transection in adult rats, followed by OB-OEC transplantation. This is the only model in which axon regeneration can be differentiated from axon sparing found in incomplete SCIs. An introductory paragraph now summarizes and references data generated from these studies that specifically addresses questions about how OECs modify the injury site and facilitate axonal outgrowth into and across into the lesion core. While relatively few axons cross the entire injury site to reach the caudal spinal cord, many more axons project into the injury site of OEC-transplanted rats compared to those in control rats. Quantification of axonal outgrowth into the lesion site of completely transected, OEC-transplanted rats from three previous long-term studies is now discussed in the Introduction. Based on both physiological and anatomical evidence reviewed from our previous work, we hope the editors and Reviewer agree that our previous studies have shown that OECs promote axonal outgrowth and modify the injury site.

Page 5, Introduction:

“Together with collaborators, we conducted six spinal cord injury studies in adult rats with a completely transected, thoracic spinal cord model followed by OB-OEC transplantation (Kubasak et al., 2008; Takeoka et al., 2011; Ziegler et al., 2011; Khankan et al., 2016; Thornton et al., 2018; Dixie, 2019). Results from five of our six studies showed physiological and anatomical evidence of axonal regeneration into and occasionally across the injury site. In 6-8-month-long studies, Takeoka et al. (2011) and Ziegler et al. (2011) reported physiological evidence of motor connectivity across the transection in OEC- but not media-transplanted rats. These experiments used transcranial electric stimulation of the motor cortex or brainstem to detect motor-evoked potentials (MEPs) with EMG electrodes in hindlimb muscles at 4- and 7-months post-transection. After 7 months, 70% of OEC-treated rats responded to stimulation with hindlimb MEPs (motor cortex, 5/20; brainstem 12/20; Takeoka et al, 2011). A complete re-transection above the original transection was carried out one month later and all MEPs in OEC-injected rats were eliminated. These results provide physiological evidence of axon conductivity across the injury site in OEC-treated rats. Additionally, three of our long-term studies evaluated anatomical axonal outgrowth of the descending serotonergic Raphespinal pathway into and through the injury site. Significantly more serotonergic-labeled axons crossed the rostral inhibitory scar border (Takeoka et al., 2011) or occupied a larger area within the injury site core (Thornton et al., 2018, Dixie, 2019) in OEC-transplanted rats than in fibroblast or media controls. In addition, significantly more neurofilament-labeled axons were found within the lesion core of OEC-transplanted versus control rats (Thornton et al., 2018, Dixie, 2019).”

Page 7, Results: We revised the sentence below and added additional information.

“We note, however, that such bridge formation is rare following severe spinal cord injury in adult mammals and was detected in 2 out of 8 OEC-transplanted rats and 0/11 media or fibroblast-transplanted controls in this study (Dixie, 2019). Combined with the 1/5 OEC-transplanted rats with axons crossing the injury and 0/5 fibroblast controls in our previous study (Thornton, 2018), we observed bridges in 3/13 OEC-transplanted rats vs 0/16 controls (p=0.042, two-sample proportion test). Bridge formation, in conjunction with the additional physiological and anatomical evidence of axonal connections across the injury site presented in our previous studies, strongly supports the capacity of OECs in neural repair.”

Page 46, Figure legend 1: We added statistical data to the legend

“Bridge formation across the injury site was observed in 2 of 8 OEC-transplanted and 0 of 11fibroblast- or media-transplanted spinal cord transected rats. Combined with the 1/5 OEC-transplanted rats with axons crossing the injury and 0/5 fibroblast controls in our previous study (Thornton, 2018), we observed bridges in 3/13 OEC-transplanted rats vs 0/16 controls (p=0.042, two-sample proportion test).”

(2) The additional discussion of sex differences in OEC bridging elaborates on the choice to study female rats, citing bladder challenges in male rats, but does not note salient clinical implications of this choice. Men account for ~80% of spinal cord injuries and likely also have worsened urinary tract issues, so it would be important to acknowledge this clinical fact and consider including males in future studies.

Response: We agree that studying SCI repair in male rodents is very important as most people with these injuries are male. We did find one publication by Walker et al. (2019, Journal of Neurotrauma 36:1974-1984) that looked at sex differences in aged-matched male and female rats after a moderate contusion SCI. They examined a number of histological and functional features, and did not find many differences between the genders. Compared to studies of moderate SCI, studies using a completely transected spinal cord model must carry out manual bladder expressions a minimum of twice a day throughout the entire 5 to 7-month study in order to maintain kidney health. Because male urethras are much longer than those of females, males are much more likely that females to die from kidney disease during a complicated, long-term studies such as ours. Fortunately, most SCIs in humans are contusions rather than complete transections so an incomplete contusion model is most appropriate for studying sex differences. We modified the previous statement in our Discussion section as below.

Page 25, Discussion

“We acknowledge that in humans, males account for ~80% of spinal cord injuries (National Spinal Cord Injury Statistical Center, 2024) and sustain more serious urinary tract issues than females. We examined females in the current study due to practical experimental considerations, but it is necessary to examine males in future studies.”

**Recommendations for the authors:**

**Reviewer #2 (Recommendations for the authors):**
(1) It is strongly recommended that some sort of quantification of bridging be included in the figures or in a table, whether this is the number of rats showing bridges, the percent area of OECs near the lesion site, or some other meaningful analysis.As discussed in the response in Challenge section (1) above, we observed bridges in 3/13 OEC transplanted rats vs 0/16 controls across our two most recent studies. In addition, we added evidence of physiological and anatomical axonal connections across the injury site from our previous studies. We have added the additional information in the Introduction, Results, and Figure legend 1.(2) It is recommended that clinical sex differences in spinal cord injury (with ~80% occurring in men) be acknowledged in the Discussion. This clinical fact could be directly mentioned without much justification.

See Challenge (2) above and addition to the Discussion on page 25.

(3) Figs. 1, 5, 6: There is still no quantification included for these figures, which detracts from the ability of readers to understand the context and importance of these results. It is recommended to include quantification for these figures.

Response regarding quantification associated with Figures 1, 5 and 6:

Regarding Figure 1: We have discussed the additions to the text of the Introduction, Results and the legend of Figure 1 in detail on pages 2-3 of this response. These are important new additions to our paper.

Regarding Figure 5: We added quantitative information regarding the analysis of Connective Tissue Growth Factor (Ctgf) expression in the injury site.

Page 10-11, Results:

“We found high levels of Ctgf expression in GFP-OECs (n=4 rats) that bridged much of the injury site and also detected Ctgf on near-by cells (Figure 5d, d1-2). GFP-labeled fibroblast transplantations (n=3 rats) served as controls and also expressed Ctgf.”

Page 36, Methods:

“To examine Ctgf expression in the spinal cord lesion site, we processed 1 slide per animal with ~6 equally-spaced sagittal sections throughout spinal cord from the Khankan et al. (2016) study. Our aim was to assess if transplanted OECs (n=4 rats) and transplanted fibroblasts (n=3 rats) express CTGF in the injury site.”

Regarding Figure 6: The statistics for Figure 6 are found on page 13 of the Results section and page 38 of the Methods section. We now added the statistics to the Figure 6 legend on page 49.

Page 13, Results:

“To determine if the proliferative OECs differ in appearance from adult OECs, and whether there is concordance between our OEC subtypes based on gene expression markers and previously described morphology-based OEC subtyping (Franceschini & Barnett, 1996), we analyzed OECs identified with the anti-Ki67 nuclear marker and anti- Ngfr^p75^ (Figure 6g-h). Of the Ki67-positive OECs in our cultures, 24% ± 8% were strongly Ngfr^p75^-positive and spindle-shaped, whereas 76% ± 8% were flat and weakly Ngfr^p75^-labeled (n=4 cultures, *p*=0.023). Here we show that a large percentage (~3/4^ths^) of proliferative OECs are characterized by large, flat morphology and weak Ngfr^p75^ expression resembling the previously described morphology-based astrocyte-like subtype. Our results indicate the two types of OEC classifications share certain degrees of overlap, indicating similarities but also differences between the two classification methods.”

Page 38, Methods: Morphological analyses of Ki67 OEC subtypes

“To determine if OEC progenitor cells marked with Ki67 immunoreactivity have a distinctive morphology, purified and fixed OEC cultures from 4 rats were processed with anti- Ngfr^p75^, anti-Ki67 and counterstained with Hoechst (Bis-benzimide, 1:500, Sigma-Aldrich, #B2261). Images were acquired from 7-10 randomly selected fields/sample using an Olympus AX70 microscope and Zen image processing and analysis software (Carl Zeiss). We distinguished the larger, flat ‘astrocyte-like’ OECs from the smaller, fusiform ‘Schwann cell-like’ OECs, and recorded their expression of Ngfr^p75^ and Ki67. Cell counts from each field were averaged per rat and then averaged into a group mean ± SEM. A Student t-test was conducted to compare the effect of Ngfr^p75^-labeled cell morphology and the proliferative marker Ki67. Statistical significance was determined by *p* < 0.05.”

Page 49, Figure 6 legend:

“Of the OEC progenitors that express Ki67, 76% ± 8 of them display low levels of Ngfr^p75^ immunoreactivity and a “flat” morphology (g2, h2; green nuclei, arrowheads). The remainder of Ki67-expressing OECs express high levels of Ngfr^p75^ and are fusiform in shape (24% ± 8%, n=4 cultures, Student-t test, p=0.023).”

(4) Fig. 9: Quantification is still not included in the figure for these Western blots, although it is appreciated that the authors included some quantification in their response letter. Including this in the figure would provide clarification for the reader.

Thank you for your suggestion. We now add the quantification to figure 9, together with the methods used for western blot quantification and the figure legend.

Page 32, Methods:

“For quantification, ImageJ software (NIH) was used to analyze the densitometric data. Western blot images at 400, 300, and 150 kDa resolution were converted to grayscale followed by manually defining a Region of Interest (ROI) frame that captured the entire band in each lane using the "Rectangular" tool. The area of each selected band was measured by employing the same ROI frame around the band to record the integrated density, “Grey Mean Value”. Background measurements were similarly quantified, and background subtraction was performed by deducting the inverted background from the inverted band value. For relative quantification, target protein bands were normalized to the corresponding loading control (GAPDH) to derive normalized protein expression (fold change). Band intensities were quantified in triplicate for each sample. Data were analyzed with the Mann-Whitney U test to compare normalized protein expression between the *Reln-/-* group and the other groups. A one-sided p-value was calculated to test the hypothesis that protein expression levels in the other groups are greater than those in the *Reln-/-* group (negative control). Statistical significance was determined at p < 0.05. Analysis was performed using GraphPad Prism (version 9).”

Page 52, Figure legend 9f:

“(f) Quantitation of multiple isoforms of Reelin from 4-15% gradient gels. Positive and negative controls are *Reln+/+* and *Reln-/-* mouse cortices. Both rat tissue from the ONL (n=3) and CM (n=9) contain more 400 and 300 kDa Reelin compared to the *Reln-/-* mouse. Bars represent the standard deviation of the mean. One-sided Mann-Whitney U test was used to test that protein expression levels in the other groups are greater than those in the *Reln-/-* group, indicative of significant expression of *Reln* in the test groups. *p < 0.05.”